# Fast delivery of heralded atom-photon quantum correlation over 12 km fiber through multiplexing enhancement

Sheng Zhang [1,3], Jixuan Shi[1,3], Yibo Liang[1], Yuedong Sun[1], Yukai Wu[1,2], Luming Duan [1,2] ✉ & Yunfei Pu [1,2] ✉

Distributing quantum entanglement between distant parties is a significant but difficult task in quantum information science, as it can enable numerous applications but suffers from exponential decay in the quantum channel. Quantum repeaters are one of the most promising approaches towards this goal. In a quantum repeater protocol, it is essential that the entanglement generation speed within each elementary link is faster than the memory decoherence rate, and this stringent requirement has not been implemented over a fiber of metropolitan scale so far. As a step towards this challenging goal, in this work we experimentally realize multiplexing-enhanced generation of heralded atom-photon quantum correlation over a 12 km fiber. We successively generate 280 pairs of atom-photon quantum correlations with a train of photonic time-bin pulses filling the long fiber, and read out the excited memory modes on demand with either fixed or variable storage time after successful heralding. With the multiplexing enhancement, the heralding rate of atom-photon correlation can reach 1.95 kHz, and the ratio between the quantum correlation generation rate to memory decoherence rate can be improved to 0.46 for a fiber length of 12 km. This work therefore constitutes an important step towards the realization of a large-scale quantum repeater network.

Quantum repeaters are most promising way to distribute quantum entanglement between two distant locations[1–3], which can then be used for various applications such as quantum communication[4–6], networked quantum sensing[7,8], and distributed quantum computing[9]. Recently, many achievements have been realized in this direction. Heralded entanglement generation in an elementary link of quantum repeater has been realized at both short and long lengths[6,10–24]. Memory-enhanced connection of two atom-photon entanglements[25,26] and 3-node quantum network[18,24] have also been implemented recently. Besides, there are also many achievements following the approach of memory-less quantum repeater[27–29].

A critical requirement for the scaling of a quantum repeater is that the expected time cost of entanglement generation in each elementary link should be shorter than the memory coherence time[11,13], to enable the synchronization and connection between asynchronously entangled elementary links. This requirement sets a stringent criterion for the performance of quantum memory in both coherence time and the ability to interface atom-photon entanglement or correlation with high efficiency. A figure of merit to quantitatively characterize the ability to deliver quantum entanglement within memory coherence time is the quantum link efficiency $\eta_{\text{link}}$ defined in[13], which can be decomposed as:

$$\eta_{\text{link}} = \frac{T_{\text{coh}}}{T_{\text{ent}}} = \frac{T_{\text{coh}}}{\frac{2L}{cNp}} = \frac{c}{2L} T_{\text{coh}} pN \qquad (1)$$

where $T_{\text{coh}}$ is the memory coherence time, $T_{\text{ent}}$ is the expected time cost of generating an atom-atom entanglement in an elementary link, $L$ is the distance between the memory and the center detection station

[1]Center for Quantum Information, IIIS, Tsinghua University, Beijing 100084, PR China. [2]Hefei National Laboratory, Hefei 230088, PR China. [3]These authors contributed equally: Sheng Zhang, Jixuan Shi. ✉e-mail: lmduan@tsinghua.edu.cn; puyf@tsinghua.edu.cn

(half length of the elementary link, see Fig. 1a, b), $c$ is the speed of light in fiber, $N$ is the effective enhancement through multiplexing, and $p$ is the success probability of each trial (Note that here $p = p_c \eta$, where $p_c$ is the intrinsic excitation probability in Eq. (1) of ref. 2, and $\eta$ is the overall efficiency of the channel and detection). Another interpretation of the multiplexing enhancement $N$ is that, with the multi-mode quantum memory, one can replace the remote entanglement generated in an earlier time with a newly generated one, which effectively prolongs the $T_{coh}$ by $N$ times, and can yield the same result as in Eq. (1). It is argued that the threshold for deterministic delivery of quantum entanglement is $\eta_{link} \gtrsim 0.83$[13]. (Intuitively, with $\eta_{link} = 1$, one can yield an entanglement with high probability during the coherence time, but suffers from significant decoherence before the final output, thus an averaged fidelity of slightly higher than the threshold 0.5 can be achieved. Therefore, to achieve an average fidelity of 0.5, one needs a slightly lower link efficiency, which is $\eta_{link} \approx 0.83$.) It is also noteworthy that $\eta_{link}$ is twice the expected number of heralded atom-photon correlation generated within memory coherence time over $L$, if single-photon interference is used[2,3,13,15,18,24,30] (by receiving clicks from both detectors at the detection station, see Fig. 1a and Supplementary Note 11). As illustrated in Eq. (1), it is crucial to improve $N$, $p$, and $T_{coh}$ to achieve a high $\eta_{link}$.

This requirement has already been fulfilled on a laboratory scale (~10 m)[11,13,24], but is yet to be fulfilled if the elementary link reaches metropolitan size ($L > 10$ km), due to significantly lower repetition rate and success probability for longer distance. As the flying qubit and heralding signal need to experience a round-trip travel of $2L/c$ (>100 μs, see Fig. 1a, b) to herald the success of the entanglement generation in each excitation trial, the repetition rate is limited to < 10 kHz in the entanglement generation stage. To achieve a faster repetition, thus higher entanglement generation rate, multiplexed quantum repeater is proposed and implemented[3,15,16,31–36]. A significant enhancement of 62 modes in a 50-m fiber has been demonstrated with rare earth ion ensemble recently[15], while the coherence time (~25 μs) needs to be improved for longer fiber length. Another approach is to improve the success probability $p$ in each trial, and efforts on cavity enhancement[12,21,22,25,34,37] and Rydberg blockade[38] have been demonstrated recently towards this goal. The cavity-enhanced trapped ion system can generate about 0.35 expected atom-photon entanglement within memory coherence over 25 km fiber[25]. Meanwhile, issues like unwanted spontaneous emission and cavity jitter limit the indistinguishability of the heralding photon[21,37], which needs to be suppressed in the future to guarantee high-quality ion-ion entanglement without harming the efficiency significantly[21]. So far, the quantum link efficiency $\eta_{link}$ is still below 0.01 for heralded entanglement generation at a fiber length over 1 km[18–20,22,23], to the best of our knowledge.

In this work, as a step towards this important goal, we achieve fast delivery of heralded atom-photon quantum correlation with a fiber length of 12 km via a multiplexing-enhanced quantum repeater protocol. The generation rate is enhanced by 280 time-bin modes and can reach 0.46 expected remote atom-photon quantum correlation within memory coherence time. As a potential experiment in the future, it is possible to achieve a doubled link efficiency if two such heralded atom-photon quantum correlations are employed to generate a 24 km atom-atom entanglement via single-photon interference[2,3,13,15,18,24,30] (in the literal meaning of $\eta_{link}$, see Eq. (20–22) in Supplementary Note 11 for details), which will be close to the scale-up requirement to build a multi-node quantum repeater. However, the future deterministic entanglement delivery task[13] cannot be achieved under current memory performance. Necessary improvements in memory performance are needed to realize this task in the future, as discussed later in this paper. (The scheme to combine two such atom-photon quantum correlations into an atom-atom entanglement is discussed in Supplementary Note 8–11, and the necessary techniques such as deployed fiber, phase stabilization of long fiber, and synchronization of control systems have been demonstrated in previous works[18–23,23,36].) The distribution rate of remote quantum correlation equals 1.95 kHz during each 240 μs protocol (with MOT loading and pumping time excluded), or an averaged rate of 187 Hz if all time costs are counted. Furthermore, the excited memory modes can be retrieved into idler photons with either fixed or variable storage time on demand, which is suitable for applications such as the synchronization and connection of two asynchronously entangled quantum repeater segments. It is also noteworthy that the built-in random-access addressing ability in our memory can support nearly arbitrary quantum network protocols[39,40]. Finally, the signal modes are detected after transmission in the long fiber, and the classical heralding TTLs also travel in another long fiber to reach the memory, which closely resembles the real-world protocol in a long elementary repeater link. With all these achievements, this work constitutes an important step towards a long-distance quantum repeater.

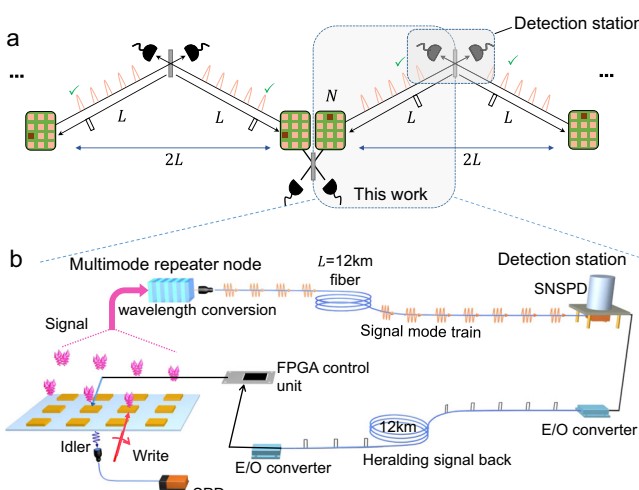

**Fig. 1 | Experimental setup and scheme. a** The position of this work in a multiplexed quantum repeater protocol. This experiment represents part of the heralded entanglement generation in a long elementary link of a multiplexed quantum repeater. With two such remote atom-photon quantum correlations, together with the phase stability techniques, deployed fiber, and the synchronization of control systems demonstrated in recent works[18–23,36], the heralded atom-atom entanglement over a metropolitan-scale elementary link can be established via single-photon interference in the future. **b** The protocol of this experiment. We use a full protocol in which the heralding signal arrives $\frac{2L}{c} = 120$ μs after the excitation to simulate the real-world application. Here in this experiment, we implement the atom-photon quantum correlation between a quantum repeater node and a signal photon on one side in an elementary link, without the interference of the photons from both sides on the beamsplitter in the detection station for future atom-atom entanglement. To measure the quantum correlation of one side, the beamsplitter is not necessary. Thus, we can replace the beamsplitter and two detectors with one detector as depicted in **b**. In this protocol, pairs of quantum correlations between signal photon modes and corresponding memory (spin-wave) modes are generated successively. The signal modes are converted to C band and sent into a 12 km fiber one by one in time-bin pulses and detected after fiber transmission. The successful detection of a signal photon is converted to TTL pulse at the detector and is further sent back to the memory through another 12 km fiber with the help of two E/O (electric-optical) converters. The memory receives the heralding TTL pulse and reads out the corresponding memory mode according to the arrival time of the TTL pulse. The detection station and the memory are 5 m apart in a lab. The fibers are also in the same lab.

## results

We implement the multiplexed DLCZ (Duan-Lukin-Cirac-Zoller) quantum memory by combining spatial[39–41] and angular[33,42] dimensions. We first realize a $10 \times 10$ spatially multiplexed memory array via 2D AOD addressing units[39,40], as shown in Fig. 2a. Here each memory

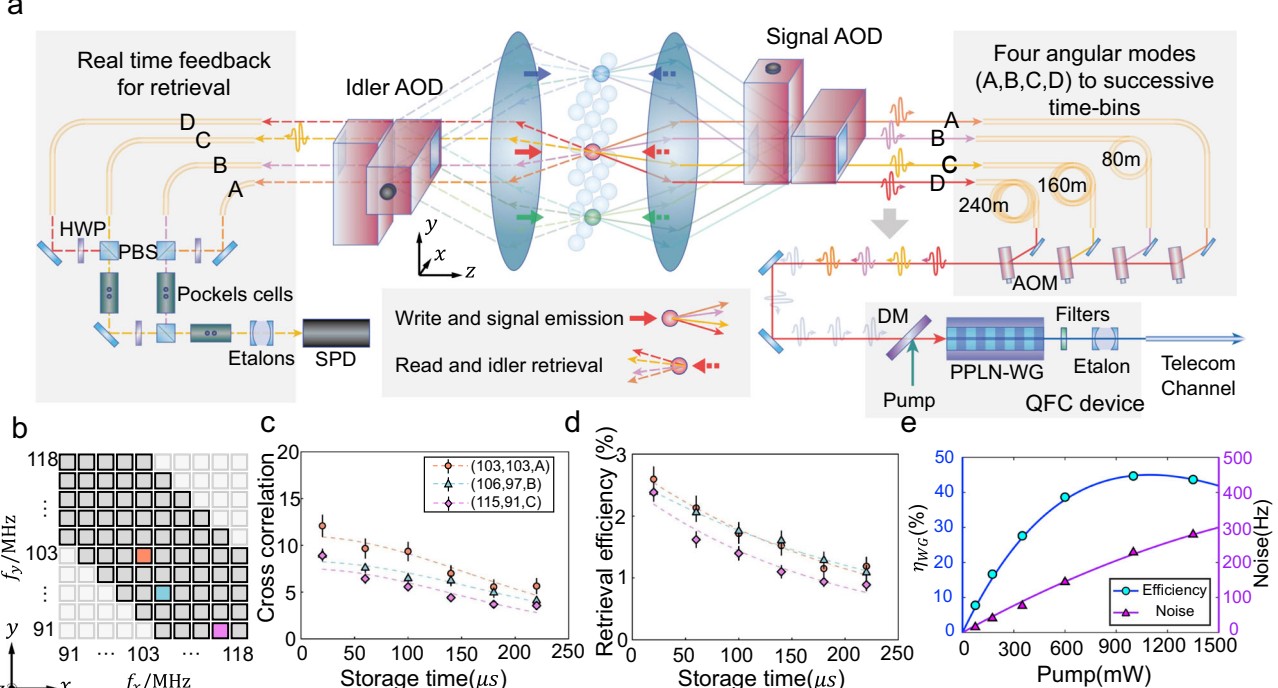

**Fig. 2 | Building a multiplexed quantum repeater node with 280 modes. a** The detailed experimental setup. We combine 70 spatial and 4 angular dimensions into a total of 280 time-bin modes. Here each combination of the spatial and angular modes can be individually addressed by the 2D AOD system with the help of AOMs (acoustic-optical modulators) and EOMs (electrical-optical modulators). The signal mode at 795 nm is converted to 1546 nm on a PPLN (Periodically poled lithium niobate) waveguide. SPD represents a single-photon detector. **b** The corresponding AOD frequencies for each memory cell. **c, d** The performance of the memory array. We choose 3 memory modes with each locating at different representative regions of the memory array and emitting at different emission angles (the orange circle represents the memory mode with $f_x = 103$ MHz, $f_y = 103$ MHz, emission angle in A

direction as shown in **a**, **b**; the blue triangle and the purple diamond represents (106, 97, B) and (115, 91, C), respectively, and demonstrate corresponding retrieval efficiency (including all losses) and cross-correlation $g_{s,i} = \frac{p_{s,i}}{p_s p_i}$ in each mode with varying storage time. The corresponding storage time of the three modes are $227 \pm 21$ μs, $261 \pm 22$ μs, and $190 \pm 22$ μs, respectively. **e** The performance of wavelength conversion. The device efficiency (circle) of the PPLN waveguide and the noise counts received by the detector (triangle) are measured under different pumping powers. The highest device efficiency is ~45% with a pumping laser ~1 W, and the end-to-end efficiency is about 12% due to multiple stages of filtering and fiber connections. Error bars represent one standard deviation in this figure.

cell is a micro-ensemble of Rb[87] atoms, which is different part of a single free-expanding atomic cloud, and these 100 cells are not loaded into optical trap arrays. All the cells are prepared via an optical pumping first, which pumps the atoms to $|5S_{1/2}, F = 1, m_F = +1\rangle$, followed by a microwave $\pi$-pulse to transfer the population to $|5S_{1/2}, F = 2, m_F = 0\rangle$ (see supplementary information). The detailed protocol of the excitation and heralding of atom-photon quantum correlation can be found later in this paper. The time overhead inlcude MOT loading time of 22 ms, and a 60 μs optical pumping before each of the ten rounds of the 240 μs excitation and heralding. We excite 70 memory cells one by one via the individually addressed write beam with a switching time of 1.7 μs (see Fig. 1b), which drives spontaneous Raman transition to create the quantum correlation between signal photon at 795 nm and collective atomic excitation (spin-wave) based on two clock states $|g\rangle \equiv |5S_{1/2}, F = 2, m_F = 0\rangle$ and $|s\rangle \equiv |5S_{1/2}, F = 1, m_F = 0\rangle$ following the DLCZ protocol[2,26]. Here four emission angles of signal photon are collected in a single write process, and quantum correlations are generated in these four pairs of signal modes and corresponding spin-wavemodes[33,42] during the excitation of each memory cell. Each of the four angular signal modes has an angle of 0.3° to the write beam, and the averaged coherence time for each spin-wave mode is 235 μs.

After the excitation, the 4 angular modes of signal photon go through the signal AOD simultaneously but are separated spatially (see Fig. 2a and supplementary information). We further pick out each signal mode after AOD and send them into delay fibers with different lengths, then combine them into 4 successive time-bin modes in a single fiber

with a time interval of 400 ns. In this way, we can generate quantum correlations between 280 time-bin signal modes and corresponding spin-wave modes by successively exciting the 70 memory cells in the array, with the 280 signal modes entering the same fiber one by one. The corresponding AOD frequency $f_x$ and $f_y$ for addressing each memory cell is illustrated in Fig. 2b. Note that as we keep the detuning of the write beam frequency to a constant value at the location of the atoms while using the write AOD to scan the array, the frequency of signal photon sent into the long fiber can be varied depending on $f_x + f_y$ for different memory cells. In order to minimize the inhomogeneity in the transmission efficiency for signal photons from each mode as the full bandwidth of the etalon in the frequency conversion is only 70 MHz, we select the 70 cells out of the 100-cell array as shown in Fig. 2b, to minimize the max difference in $f_x + f_y$ to 28 MHz. It is noteworthy that this frequency variation due to AOD scanning will not influence the single-photon interference to generate a heralded atomatom entanglement in the future, as analyzed in Supplementary Note 8. The time-bin signal modes at 795 nm further go through a PPLN (Periodically poled lithium niobate) waveguide and are converted to 1546 nm via DFG (Difference frequency generation)[18,19,22,25,43–46]. The conversion efficiency and rate of noise counts are illustrated in Fig. 2e. Note that the SNR will increase with higher excitation probability of signal photon (Supplementary Fig. 7). The time-bin train of converted signal modes are further sent through the 12 km fiber to the detection station (~5 m away from the memory), where the signal modes are converted to TTL (Transistor-Transistor Logic) pulses at the SNSPD (superconducting nanowire single-photon detector).

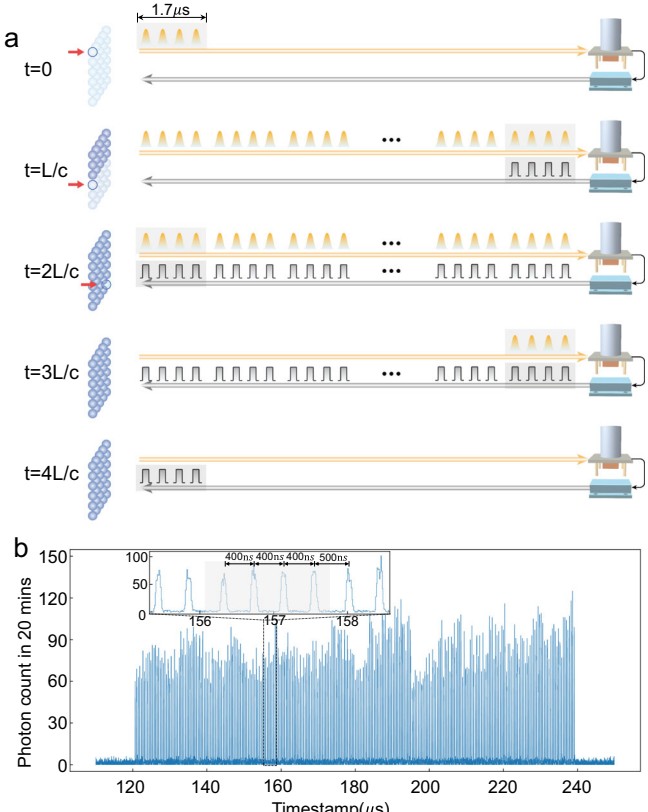

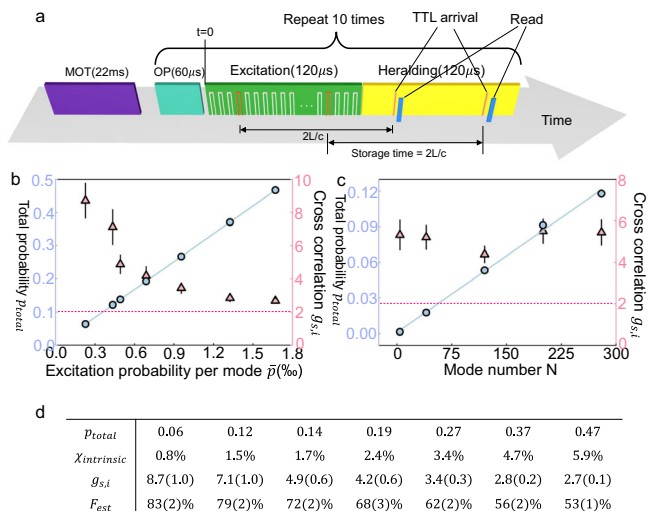

| $p_{total}$ | 0.06 | 0.12 | 0.14 | 0.19 | 0.27 | 0.37 | 0.47 |
|---|---|---|---|---|---|---|---|
| $\chi_{intrinsic}$ | 0.8% | 1.5% | 1.7% | 2.4% | 3.4% | 4.7% | 5.9% |
| $g_{s,i}$ | 8.7(1.0) | 7.1(1.0) | 4.9(0.6) | 4.2(0.6) | 3.4(0.3) | 2.8(0.2) | 2.7(0.1) |
| $F_{est}$ | 83(2)% | 79(2)% | 72(2)% | 68(3)% | 62(2)% | 56(2)% | 53(1)% |

**Fig. 3 | Remote heralding with 280 time-bin modes. a** The heralding protocol of remote atom-photon quantum correlation over an $L = 12$ km fiber enhanced by 280 time-bin modes. The first four modes are generated at $t = 0$ when the addressed write beam excites the first memory cell. The time-bin modes of a signal photon are traveling in the long fiber as a long pulse train, with an interval at 400 ns (between modes from the same cell) or 500 ns (between the 4th mode from the previous cell and the 1st mode from the current cell). **b** The histogram of the arrival time for the heralding TTL received by the memory, at an average success probability of $\bar{p} = 0.18\%$. The inset is a zoom-in in the long time-bin pulse train, the shaded four modes are from the same memory cell. The temporal width of each signal mode is ~100 ns. The bin size of the histogram is 10 ns.

**Fig. 4 | The protocol and results when the storage time is fixed. a** The protocol with the storage time fixed during the retrieval. 10 rounds of repetitions lasting ~3 ms follow the 22 ms MOT loading stage. In each round, the generation of heralded quantum correlation starts after the 60 µs optical pumping. During the excitation stage, we start to excite the first memory cell at $t = 0$, and it costs ~120 µs to excite all the 280 modes. The heralding stage lasts from $t = 120$ µs to $t = 240$ µs. Once a heralding TTL pulse is received, the memory immediately read the corresponding spin-wave mode out. In this case, the storage time is always fixed to 130 µs. The memory can register and retrieve multiple heralded quantum correlations (at most 3) in each round. **b**, **c** The total probability $p_{total}$ (blue circle) and cross-correlation $g_{s,i} = \frac{p_{s,i}}{p_s p_i}$ (pink triangle) with varying average success probability $\bar{p}$ and mode number $N$. The dashed line represents the upper bound of classical correlation at 2. Here $N = 280$ in **b** and $\bar{p} = 0.42‰$ in **c**. **d** Here, we list the total excitation success probability $p_{total}$, intrinsic excitation probability $\chi$ in the excitation of each mode, cross-correlation $g_{s,i}$, and the estimated atom-atom entanglement fidelity if two such atom-photon quantum correlation is exploited to generate an atom-atom entanglement in the future, for the seven data points demonstrated in **b**. Error bars represent one standard deviation in this figure.

To notify the memory, the heralding TTL pulses are converted to optical pulses by an E/O converter and sent back to the memory via another 12 km fiber, as shown in Figs. 1b and 3a. After the optical pulses arrive at the memory, they are converted back to TTL pulses via another E/O converter. The FPGA control unit gets the heralding signal and identifies which spin-wave mode (both spatial and angular) is excited depending on the TTL arrival time, and reads out the corresponding spin-wave mode to an idler photon, which can be used for verification of the distant quantum correlation or further connection of neighboring repeater segments. This protocol demonstrated here simulates a real-world protocol for heralded generation of distant atom-photon quantum correlation, which can be further employed to generate atom-atom entanglement in an elementary repeater link via single-photon interference[2,3,15,18,30]. The scheme for this is discussed in Supplementary Notes 8–11 in detail.

When 280 signal modes are used, we need $70 \times 1.7 = 119$ µs to finish the excitation process, which approximately corresponds to $\frac{2L}{c} = 120 \mu$ s, the round-trip travel time in both of the 12 km fibers for qubit and TTL. This means the heralding TTL pulse for the first signal mode arrives at the memory right after the last memory cell is excited, as illustrated in Fig. 3a. The 12 km qubit fiber is fully filled from $t = \frac{L}{c}$ to $\frac{2L}{c}$, the control unit starts receiving the heralding TTL since $t = \frac{2L}{c}$, and this process keeps going until $t = \frac{4L}{c} = 240 \mu$ s. The histogram for the arrival time of heralding TTLs is registered and illustrated in Fig. 3b,

which also represents the success probability of each signal modes. Here we can execute 280 trials in each 240 µs round, with an effective repetition rate at 1.17 MHz, which is >100 times faster than the maximum repetition rate $\frac{c}{2L} = 8.3$ kHz if there's no multiplexing enhancement.

After the successful heralding in any of the 280 modes, we read out the stored spin-wave mode to demonstrate the retrieval of the quantum information stored in the quantum memory in two different styles, which are (i) reading out the corresponding memory mode immediately after receiving a heralding TTL, and (ii) reading out the corresponding memory mode at a pre-defined time no matter when the heralding TTL is received. It is also noteworthy that our memory can perform an arbitrary protocol in the reading as it can be performed in a random-access way[39,40], not limited to these two styles demonstrated here.

In the first read-out style, the storage time is fixed, but the read-out time $t$ is random (read-out time $t$ is the time of executing the read operation, with $t = 0$ represents the beginning of the excitation stage, see Fig. 4a). As illustrated in Fig. 4a, the control unit will identify and read out the excited spin-wave mode immediately after receiving a heralding TTL in the 120 µs heralding stage. This is achieved by dynamically controlling the RF (radio frequency) signal sent to the AOD depending on the TTL arrival time, which can be completed in 10 µs after receiving the heralding TTL (see supplementary information). Thus the storage time is always $\frac{2L}{c} + 10 = 130 \mu$ s for each of the 280 modes. If more than one TTLs are received in the heralding stage, we can either read all of them out or immediately terminate the heralding stage right after the first heralding TTL is received. Here we

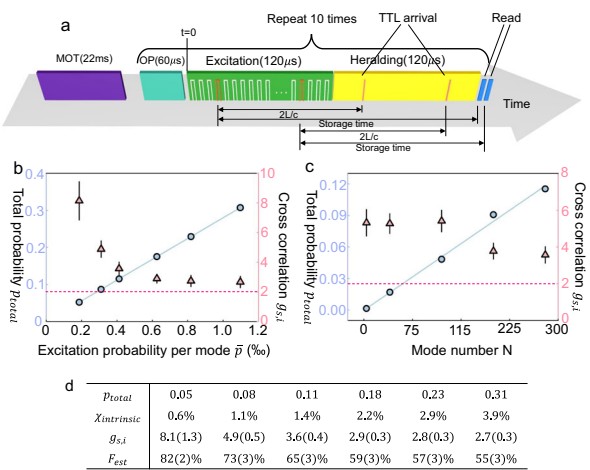

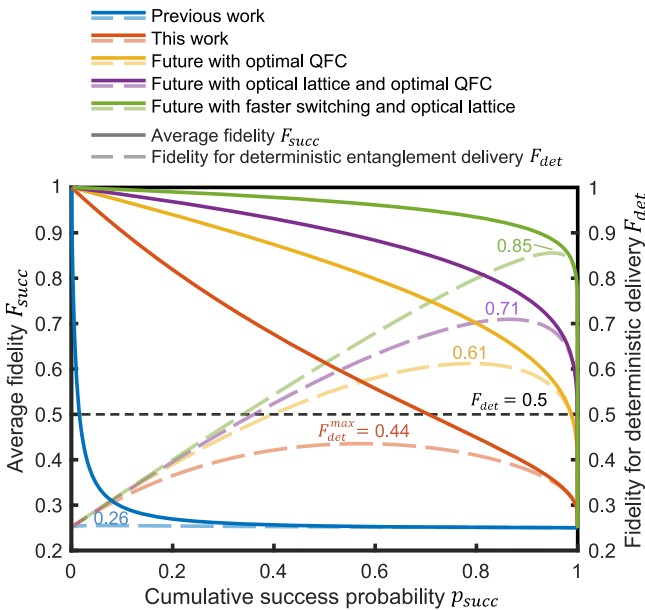

**Fig. 5 | The protocol and results when the retrieval time is user-defined. a** The protocol where the retrieval time is fixed to the end of the heralding stage. In the heralding stage, once a heralding TTL pulse is received, the memory registers this event, waits, and reads out the corresponding spin-wave mode at a pre-defined time (250 μs here). In this case, the storage time is random over the range from 130 μs to 250 μs. The protocol can also register and retrieve multiple heralded quantum correlations (at most 3). **b, c** The total probability $p_\text{total}$ and cross-correlation $g_{s,i}$ with varying average success probability $\bar{p}$ and mode number $N$. The dashed line represents the upper bound of classical correlation at 2. Here $N = 280$ in b and $\bar{p} = 0.43‰$ in **c**. **d**, Here we list the total excitation success probability $p_\text{total}$, intrinsic excitation probability $\chi$ in the excitation of each mode, cross-correlation $g_{s,i}$, and the estimated atom–atom entanglement fidelity if two such atom-photon quantum correlation is exploited to generate an atom-atom entanglement in the future, for the six data points demonstrated in **b**. Error bars represent one standard deviation in this figure.

**Fig. 6 | Fidelity and success probability for atom-atom entanglement in the future.** We simulate the average fidelity when the heralding is successful ($F_\text{succ}$), the fidelity for deterministic entanglement delivery ($F_\text{det} = p_\text{succ} F_\text{succ} + (1 − p_\text{succ})F_\text{unent}$, with $F_\text{unent} = 0.25$, see ref. 13) versus the cumulative success probability ($p_\text{succ}$) for the task of generating heralded atom-atom entanglement over 24 km fiber by connecting two atom-photon quantum correlation over 12 km fiber through single-photon interference. The protocol is still the 120 μs excitation followed by 120 μs heralding used in the atom-photon quantum correlation generation in this work. Here we analyze the relation between fidelity and success probability under five different quantum memory performances, including the memory performances in: (i, blue curve) a previous work with low coherence time and slow switching speed[48], (ii, brown curve) this work, (iii, yellow curve) based on this work, but we assume the efficiency of frequency conversion is improved to the current best in the future[19], (iv, purple curve) loading the atomic cloud into two-dimensional optical lattice array and the optimal frequency conversion efficiency in the future, and (v, green curve) loading into optical lattice array, with optimal frequency conversion, and improve the switching speed by four times in the future. The cumulative success probability is varied by adjusting the excitation probability in each mode. It is shown that the future deterministic entanglement delivery task cannot be achieved under current memory performance as the maximally achievable fidelity $F_\text{det}^\text{max} = 0.44$ is smaller than the threshold 0.5. Deterministic entanglement delivery ($F_\text{det}^\text{max} > 0.5$) can be achieved with necessary improvements in the future with improved frequency conversion or optical lattice array.

choose to read out at most 3 excited spin-wave modes during each heralding stage for accelerated data collection. A detailed explanation of why the data collection is accelerated can be found in Supplementary Note 10. Idler photon from different spin-wave modes are also combined into a single fiber via an EOM (electro-optical modulator) network, as shown in Fig. 2a. A detailed explanation of why the data collection is accelerated can be found in Supplementary Note 10.

Here we use $p_\text{total} = \sum_{i=1}^{N} p_i$ to characterize the total success probability in one round of excitation, where $p_i$ is the success probability of detecting the $i$th signal mode at the detection station (and $i$th heralding TTL at the memory), and $N$ is the number of total used modes. We also use the cross-correlation $g_{s,i} = \frac{p_{s,i}}{p_s p_i}$ to characterize the quality of the heralded atom-photon quantum correlation (see supplementary information for details). In the DLCZ quantum repeater protocol, the efficiency and the quality are in a trade-off[2], and here we demonstrate the total efficiency $p_\text{total}$ and cross-correlation $g_{s,i}$ with a varying average success probability $\bar{p} = \frac{1}{N} p_\text{total}$ in Fig. 4b. As shown in Fig. 4b, the $p_\text{total}$ increases linearly with $\bar{p}$ and $g_{s,i}$ decays with $\bar{p}$. When $\bar{p} = 0.167\%$, we can achieve a total success probability as high as $p_\text{total} = 0.47$, with the cross-correlation at $g_{s,i} = 2.67 \pm 0.15$, which provides clear evidence for quantum correlation as the classical bound is 2[47]. The whole time cost of this protocol is the sum of the excitation stage 120 μs and the heralding stage 120 μs, which equals 240 μs. Here, the 60 μs optical pumping can be combined into the overhead like MOT loading because optical pumping is only needed once before the beginning of the 240 μs protocol, and no longer needed in the middle of the protocol (from $t = 0$ to $t = 240$ μs). This also means that we can generate heralded atom-photon correlation at a rate of $\frac{0.47}{240\,\mu s} = 1.95$ kHz in each round of 240 μs. If we take everything (including MOT loading and optical pumping) into account, the average remote atom-photon correlation generation rate is about 187 Hz, with a duty cycle of 9.6%. Given the

average coherence time $T_\text{coh} = 235$ μs (see Supplementary Note 1 for details), here we can generate 1.95 kHz × 235 μs = 0.46 expected atom-photon correlation within memory coherence time. We can also vary the number of modes used in this protocol, and the results with $\bar{p} = 0.042\%$ are shown in Fig. 4c. As shown in Fig. 4c, the total success probability increases linearly with $N$, which clearly illustrates the multiplexing enhancement. When $N < 280$, there will be a gap between the excitation and heralding stage, but the storage time is still fixed to 130 μs. The cells used in different $N$ are described in detail in the supplementary information.

In the second read-out style, no matter when the heralding TTL is received, the memory always reads out the stored spin-wave mode at a user-defined time $t$, and here we set this time to $t = 250$ μs at the end of the heralding stage to cover all the 280 modes, as shown in Fig. 5a. As a consequence, the storage time is variable for example, 250 μs for the 1st mode, and 130 μs for the 280th mode. This protocol is more favorable than the first protocol for the connection between different repeater segments as it simulates the synchronization and connection between two asynchronously entangled elementary links, which requires the quantum memory to handle a variable storage time and

can be retrieved on demand. Here we can also register at most 3 heralding TTLs and read corresponding spin-wave modes out for faster data collection. We also demonstrate the total heralding probability $p_{total}$ and cross-correlation $g_{s,i}$ with varying average probability $\bar{p}$ and mode number $N$ in Fig. 5b, c. We can achieve $p_{total} = 0.30$ with the quality of quantum correlation still above the classical bound, which corresponds to an equivalent quantum link efficiency $\eta_{link} = 0.60$ in this case. The slight decay of corresponding $p_{total}$ and cross-correlation compared to fixed-storage time case mainly originatess from the longer storage time in this case.

With the heralded atom-photon quantum correlation over a 12 km fiber demonstrated in this work, here we also analyze the performance of future atom-atom entanglement by interfering two such atom-photon quantum correlations via single-photon interference[2,3,13,15,18,24,30]. In DLCZ protocol, the excitation probability and the cross-correlation are in a trade-off, which means the cumulative success probability and the atom-atom entanglement fidelity are also in a trade-off. Here we simulate the entanglement fidelity versus the cumulative success probability, for the task of generating a 24 km atom-atom entanglement by interfering with a pair of 12 km atom-photon quantum correlations via single-photon interference in the middle detection station, as illustrated in Fig. 1a. Here we use the same heralding protocol in this experiment which contains a 120 μs excitation stage and a 120 μs heralding stage, using the fixed-storage time read-out mode. The relation between the atom-atom entanglement fidelity and the cross-correlation is derived and analyzed in Supplementary Note 9 in the supplementary information. We simulate the relation between the cumulative success probability $p_{succ}$, average fidelity $F_{succ}$ and the fidelity for deterministic entanglement delivery $F_{det}$ under five different memory performances, as illustrated in Fig. 6. For simplicity, here we have assumed the noise of frequency conversion is negligible in Fig. 6. We can clearly see the trade-off between the fidelity and the cumulative success probability, and the improvement in both of the fidelity and the success probability by using a quantum memory with higher performance. In Fig. 6, it is also shown that the future deterministic entanglement delivery task cannot be achieved under current memory performance. To achieve this remarkable goal, we need further improvement in frequency conversion or employing an optical lattice array.

## Discussion

In summary, we experimentally realize heralded generation of atom-photon quantum correlation over 12 km fiber, with a multiplexing-enhanced efficiency via the use of 280 modes. We demonstrate an expected delivery of 0.46 atom-photon quantum correlation within memory coherence time, which represents the current state-of-the-art. Two different read-out styles are also demonstrated in this work and the quantum correlation can be generated with a record-high rate over a long fiber. In the future, we can use deployed fibers and distant quantum repeater nodes to establish quantum repeater links between spatially separated locations[18–23,36]. There is still room to improve the efficiency of wavelength conversion (12%) considering the state-of-the-art is 57%[19], and this can further improve corresponding $\eta_{link}$ by several times. We can also improve the performance of the memory by loading the memory cells into a two-dimensional optical lattice array. With the improved storage time of ~50 ms[26], the memory can achieve an equivalent quantum link efficiency of over 100, which will constitute a promising platform for building a multi-layer quantum repeater network extended over a metropolitan or continental scale in the future.

## Data availability

The data generated in this study have been deposited in the Zenodo database under accession code https://doi.org/10.5281/zenodo.13629436.

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

## Acknowledgements
We acknowledge helpful discussions with Xiaohui Bao, Yong Yu, Jun Li, Wei Zhang, and Zhiyuan Zhou. This work is supported by the Innovation Program for Quantum Science and Technology (No. 2021ZD0301102), the Tsinghua University Initiative Scientific Research Program, the Ministry of Education of China through its fund to the IIIS, and the National Key Research and Development Program of China (2020YFA0309500). Y.P. acknowledges support from the start-up fund and the Dushi Program from Tsinghua University. Y.W. acknowledges support from the start-up fund from Tsinghua University.

## Author contributions
S.Z., J.S., Y.L., Y.S., and Y.P. carried out the experiment. L.D. and Y.P. supervised the project. S.Z., J.S., Y.W., L.D., and Y.P. contributed to the writing.

## Competing interests
The authors declare no competing interests.
