## [Transparent Peer Review file · Nature Communications]

Fast delivery of heralded atom-photon quantum correlation over 12km fiber through multiplexing enhancement

Corresponding Author: Professor Yunfei Pu

Version 0:

Reviewer comments:

Reviewer #1

(Remarks to the Author)

The manuscript by Zhang and colleagues reports on the generation of heralded atom-photon entanglement based on atomic ensembles used as quantum memories. They use the DLCZ protocol and spatial multiplexing, which is a strong expertise from the authors. The main goal is to enhance the so-called link efficiency, i.e., the ratio between entanglement generation rate and the decoherence time, towards a value closer to a unity. This is a crucial objective for quantum networks, previously demonstrated only in solid-state systems. With atomic ensemble, the achieved values have been one to two orders of magnitude lower. In this paper, the authors play on two parameters to increase this ratio: the number of modes available by combining individual memory cells (x100) and k-vector multiplexing (x4). Furthermore, to be as close as possible as real field deployment, they included frequency conversion to telecom wavelength and a 12 km fiber link. The main claim is a significant increase of the link efficiency to 0.46 for such a fiber length.

I appreciated the work and the elegant implementation. The study is also well-characterized, with interesting technical discussions in the SI. However, I do not favor its publication in Nature Communications.

Indeed, this demonstration comes at the cost of very limited cross-correlations, a crucial parameters for further applications. The reported value of 0.47 corresponds to a value of $g_{si}=2.67$. Although this is above the classical bound of 2 (even if, being so close to this value, the full Cauchy Schwartz inequality should be verified), it is well known that such a value is insufficient to enable further applications as this functioning is not enough in the single excitation regime. Most recent networking demonstrations with doped crystals, for instance, were performed in a regime with $g_{si}\gg 10$. Previous DLCZ-based demonstrations have also much higher correlations. A value close to 2 will not enable demonstrations of atom-atom entanglement or even more entanglement swapping due to the large multi-photon components. According the figures, for a g_{si} of about 10, which would be a reasonable for a proof-of-principle demonstration, the probability drops to 0.05. The claim significantly changes in this scenario.

I have additional comments that should be addressed to clarify the paper and specific claims.

- Figure 1 contains a lot of information, making it difficult to grasp the protocol and the specific techniques employed. The authors should increase the figure size, or separate it in different figures, to improve. Actually I found figure S3 clearer and easier to understand.

-The retrieval efficiency is limited to the percent level. This is very low and not compatible with quantum networks where photons need to be read out and synchronize with others. This is a strong limitation for the scalability of the scheme. The authors should address this point in greater detail (no values of optical depth are given for instance). Similarly, in the conclusion, the potential increase of the memory lifetime via the use of optical lattices is discussed. What about the achievable efficiency and vector multiplexing in that case as the memory cell will have a much reduced extension ?

-One of the technique here to increase further the multiplexing is to use k-vector multiplexing. I m curious about the effect of this technique on added noise.

-The authors increase the storage lifetime by optical pumping, further described in the SI. What is the efficiency of the process that involves microwave pulses and the final optical depth to be used for the DLCZ scheme?

Reviewer #2

(Remarks to the Author)

In this article, the authors demonstrated the non-classicality of the atom-photon cross-correlation function over a 12 km optical fiber with a quantum link efficiency of 0.46. By integrating single-photon interference using current technologies in the future, this value can be doubled, surpassing the value of 0.83 required for deterministic delivery of quantum entanglement. To enhance the quantum link efficiency, improving the entanglement sharing rate is crucial. To increase the number of trials, the authors developed a 280-mode multiplexing of the atomic cloud by combining 70 spatial modes, each with 4 angular modes. Additionally, they used quantum frequency conversion to convert 795 nm signal photons emitted from the atomic cloud into telecom photons, thereby improving the sharing rate per mode.

There is a trade-off between the success probability and the value of the cross-correlation function. By quantitatively analyzing this relationship and setting the appropriate probability, they achieved a link efficiency of 0.46 with a g_2 value of 2.67 for a fixed storage time of 130 μ s. They also performed another protocol with a user-defined retrieval time ranging from 130 μ s to 250 μ s. In this case, the link efficiency was 0.30, slightly lower than 0.46, but they discussed potential improvements, such as enhancing the frequency conversion efficiency and introducing an optical lattice array.

Their results highlight the advantages of quantum repeaters over long distances. The experimental technologies used in this study, including the multiplexed atomic cloud, read and write sequence, remote heralding system, and frequency conversion system, are highly advanced. The experiments and results are clearly presented and easy to follow for readers including the Supplemental material. Their achievement possesses sufficient novelty and will have a strong impact on a broad range of researchers, especially in the fields of quantum networks and quantum internet.

As described below, I have questions about the quantum link efficiency which is one of the most important results. Including this, I listed several comments and questions below. After they properly address them, I recommend the publication of the manuscript in Nature Communications.

Major comments:

Q1

The cross correlation of 2.67 was observed in the protocol of the fixed storage time. How much value of the fidelity to the entanglement after the entanglement swapping based on single-photon interference is expected from the observed value?

Q2

Related to Q1, quantum link efficiency in Eq.(1) does not appear to include the amount of entanglement. This may allow for a strategy where generating low-quality entanglement with high efficiency becomes advantageous. However, in Eqs.(1),(2) and Fig.1b of Ref.[13], the quantum link efficiency should be related to the average fidelity with respect to the maximally entangled state. It is natural to consider the cost of entanglement distillation. This paper seems to lack the consideration to fill this gap. The authors should clearly explain the point.

Other comments:

Q3

In Eq.(1), the enhancement factor of the success probability related to T_{ent} is considered. Is it possible to consider the enhancement of the memory time T_{coh} , for example, by including the swapping operation of the quantum state stored in the memory to fresh memory?

Q4

The link efficiency for the deterministic delivery of entanglement is 0.83, which is lower than 1. Could the authors provide an intuitive explanation for why a value less than one is sufficient? Such a description would be helpful for readers.

Q5

How are the details of the etalons used for the detection of idler photons and for frequency conversion? How are they stabilized?

Q6

In Fig. 1d, the cross-correlations of the signal and idler photons were around 10. These values seem relatively small compared with conventional experiments based on atomic clouds. What is the main cause of the degradation of the cross-correlation functions?

For Supplemental material:

Q7

On L9 and L19, references should be added to "in our works" and "in our prior work."

Q8

On L22, the authors explain the angle between the write beam and the signal photon is 0.3 degrees. Is this small angle resolved by increasing the path lengths?

If so, could the authors tell me the lengths of the photons and the write beam? If different techniques are used, this should be clarified.

Reviewer #3

(Remarks to the Author)

In this manuscript a significant technological difficulty is addressed to establish the required entanglement rate for a fully operational quantum repeater. The quantum link efficiency in the continuous picture of [1] is used as figure of merit and it is shown that the experimental realisation is close to the required value to become fully operational in terms of decoherence and 'quantum correlation' rate. Especially it is shown that two such devices will beat the required value of 0.83, which is the benchmark to get in the mean one entangled state with a Fidelity >0.5 per attempt. The protocol is the one photon DLCZ

protocol of [2], and the experimental key element to create the necessary rate is by applying on demand addressable spatial and angular multiplexing methods, all coupled to a single mode fibre and frequency converted to the telecom C-band.

The results are convincing and the figures appropriate to understand the experiment. It is remarkable that with this experiment one frequency band of a 12km long transmission fibre is temporally filled and it seems that it is possible to go beyond by a factor 100/70 by using all the 100 memory cells instead of only 70. Furthermore, the random-access readout possibility at a selectable time is a crucial step and is well demonstrated in the second readout scheme.

My main criticism in this manuscript is a missing discussion regarding the entanglement generation rate and fidelity to address the full picture of an operational quantum repeater, especially as the realization becomes closer. To obtain entanglement with this protocol, it is necessary to combine two such devices -which they call segment- to an entangled memory-memory segment in an interferometric way. Certainly, difficulties arise which should be discussed. In this regard they point out issues with another system (line 69-74: 'Meanwhile, issues like unwanted spontaneous emission and cavity jitter limit the indistinguishability of the heralding photon [21, 36], which need to be suppressed in the future to guarantee high-quality ion-ion entanglement without harming the efficiency significantly [21].') but a discussion to convert the described 'quantum correlation' into the memory-memory entanglement is missing and should be added. I see difficulties in the required interferometric stability, starting with the two excitation lasers until the overlapping of the photons, as there are many different paths involved and also the frequency conversion is included. Also, the frequency shift due to the deflectors should be discussed in this sense. Maybe the discussion should go beyond, if two memory-memory segments get combined in the random-access manner that is used in the second readout scheme. It would be also interesting if all memories can be connected, i.e. if the spatial multiplexing allows the entanglement swapping between any of the memories.

Meanwhile a new result appeared (arXiv:2406.09480v1), it may be interesting how this compares.

Further questions and suggestions:

Line 39-42:

'It is also noteworthy that eta-link is twice the expected number of heralded atom-photon correlation generated within memory coherence time over L, if Type I protocol (single photon interference) is used [2, 3, 13, 15, 18, 24, 30] (by receiving clicks from both detectors at the detection station, see Fig. 1a).'

This sentence causes a few problems to me:

- a) I don't understand the sentence: 'eta-link is twice the expected number of heralded atom-photon correlation generated within memory coherence time over L'
 - b) The terminology 'Type I protocol' is -in my opinion- not clear and can be avoided as it is not used in the rest of the manuscript. A short reminder on formula (1) of [2] and the link of $\sqrt{p_c}$ to the success probability (\bar{p}) of this manuscript would help to understand the rest of the manuscript much faster.
 - c) '(by receiving clicks from both detectors at the detection station, see Fig. 1a).'
- The detector station is in Fig. 1.b and there is only one detector in the experiment.

Line 85-88

'This corresponds to a $\eta_{\text{link}} = 0.92$ if two such heralded atom-photon quantum correlations are employed to generate a 24 km atom-atom entanglement via single photon interference [2, 3, 13, 15, 18, 24, 30] in the future, which will be close to the scale-up requirement to build a multi-node quantum repeater.'

I don't understand why eta doubles when the link gets extended at the same time. In my opinion this is only the case when two such devices are used to bridge the same distance. Can you explain that?

Line 91

'... equals to 1.95 kHz during each 240 us protocol ...'

This formulation is not clear, I guess you mean when excluding MOT loading and pumping?

Line 112

'Here each memory cell is a micro-ensemble of Rb87 atoms, ...'

It would help if you specify the trap, i.e. the realisation of the micro ensembles ('... different memory cells are different parts of a single atomic cloud ...' taken from [38], is a tweezer array used?)

First paragraph of results (line 108-130)

Refer to Fig 3a and describe the time overhead of the protocol (MOT loading and optical pumping).

line 141

I would suggest to mention the frequency shift and the memory-cell selection due to the AO-deflectors. Especially if the interference is affected.

Line 148

How is the SNR defined? And wouldn't it be better to specify the noise photons in dependence of the pump power?

Line 165-170

'This protocol demonstrated here simulates a real-world ... via single photon interference.'

Should be discussed -and further explained- in the introduction part.

Line 187-189

'After the successful heralding in any of the 280 modes, we read out the stored spin-wave mode for further applications. Here we demonstrate the retrieval of the quantum information stored in the quantum memory'

Change to: 'After the successful heralding in any of the 280 modes, we read out the stored spin-wave mode to demonstrate the retrieval of the quantum information stored in the quantum memory'

Line 198

'In the first read-out style, the storage time is fixed, but the read-out time is random.'

It is always a programmed time, so I don't understand which time is random.

Line 210-212

'Here we choose to read out at most 3 excited spin wave modes during each heralding stage for accelerated data collection.'

Why is the data collection accelerated?

Line 225

'... varying average success probability ...'

how is the probability set and which excitation probability does this correspond (and which fidelity)?

Line 243

Refer to the measurement in the supplementary material for the coherence time as this is important for η_{link} . I would suggest explaining in the supplement how this is measured.

Line 245-249

'... which also corresponds to an equivalent quantum link efficiency $\eta_{\text{link}} = 0.46 \times 2 = 0.92$ if two such setups are combined to generate heralded atom entanglement with single photon interference in the future.'

See comment to line 85-88

Line 258

'the memory always reads out the stored spin wave mode at a user-defined timestamp'

Check formulation and the term 'timestamp' is difficult to understand.

Line 261-263

'Note that in this protocol the read-out time is fixed no matter which mode is excited, meanwhile the storage time is variable (for example, the storage time is 250 us for the 1st mode, and 130 us for the 280th mode)'

Suggestion: 'As a consequence, the storage time is variable for example, 250 us for the 1st mode, and 130 us for the 280th mode.'

Line 272

'... for faster data collection.'

See comment to line 210-212

FIG. 1

- a) In Fig 1b: typo: ',E/O convertor'
- b) In Fig 1d: Can you add dimensions to the array?
- c) In Fig 1d: The label of the cross correlation graph is not explained.
- d) In Fig 1d: Instead of SNR, I would suggest showing the noise photons and add the corresponding SNR to the p values in the text.
- e) Label: 'This experiment represents half of the heralded entanglement generation ...' In my opinion, there are components missing for the necessary phase stability, in this sense it is not half of the necessary components. I suggest finding another formulation for your experiment.

FIG. 2

- a) Specify the bin size of the histogram.
- b) 'Note that the excitation probability here is higher than in Fig. 1e, thus higher signal-to-noise ratio is achieved.' This sentence can be omitted if the noise photons are specified.

Supplementary material:

Line 9

'... previous works.' : add references

In line 52

What do you understand as the feedforward process? This name confuses me, as the readout is always performed after herald detection and not 'foreseen' from prior runs.

Figure S1c

What is the origin of the crosstalk?

Figure S1d+e

- a) Specify how the coherence time is measured.
- b) '...fitted by retrieval efficiency decay.' It is not clear what this means.

[1] Humphreys, P. C., Kalb, N., Morits, J. P., Schouten, R. N., Vermeulen, R. F., Twitchen, D. J., Markham, M. & Hanson, R. Deterministic delivery of remote entanglement on a quantum network. *Nature* 558, 268-273 (2018).

[2] Duan, L.-M., Lukin, M. D., Cirac, J. I. & Zoller, P. Long distance quantum communication with atomic ensembles and linear optics. *Nature* 414, 413-418 (2001)

Version 1:

Reviewer comments:

Reviewer #1

(Remarks to the Author)

The manuscript has been largely improved and the new version of the manuscript now provides additional discussions in the main and in the SI. These discussions clarify some of the data and their limitations, providing also insights about future implementations. Given these changes, I can now recommend publication in your journal.

Reviewer #2

(Remarks to the Author)

The authors have addressed most of my comments and questions. In particular, significant revisions have been made regarding the link efficiency, which was pointed out by all reviewers. A quantitative analysis has been conducted on several different setups and parameters for generating entanglement between the remote atoms. However, I still have concerns about their statement that they achieved the link efficiency of 0.92 using the observed rate of 0.46 in their experiment related to L113-116 and L313-319.

In Fig.6, they showed the dependency of the fidelity on the success probability using experimental parameters. However,

only from the figure, it is unclear whether the link efficiency larger than 0.83 required for the deterministic delivery of entanglement can be achieved. To clarify their statement, by adding the curve for $\eta=0.83$ on the same figure similar to Fig.1b of Ref.[13], it should be shown that, in the region of $F > 0.5$, there exists a success probability which gives the fidelity of the brown curve ("This work") larger than that of the curve for $\eta=0.83$. I guess, from the curves of Fig.1b in Ref.[13] and the shape of the brown curve in their manuscript, the fidelity of the brown curve may always be smaller. If correct, I think the link efficiency larger than 0.83 was not achieved using the observed result. The authors should clarify my concern and weaken their argument if needed.

In my opinion, even if the link efficiency larger than 0.83 using the current experimental parameters has not been shown, this paper is worthy of publication after the above concern is addressed, considering their theoretical analysis, which suggests a significant improvement of the link efficiency through an improvement of QFC efficiency likely to be feasible, and the state-of-the-art quality of their experiments including the design, atomic systems and control system.

Other comments:

(1)

In Eq.5 of the supplementary information, $P_i = \chi (1+\eta(t))\eta_i$ is used. For $\eta(t)=0$, P_i is not zero. Why? Are any assumptions being made for this equation?

(2)

For Figs.S11-e and f, the labels of the secondary vertical axes are "Link efficiency."

Are they the link efficiency, not the success probability?

The figures look strange because a larger value of χ gives a smaller value of fidelity. This result implies that the contribution of the success probability improvement is dominant compared with the fidelity decrease because the link efficiency depends on the fidelity and probability. Is my understanding correct?

Reviewer #3

(Remarks to the Author)

The authors answered all my questions and added additional detailed information in the supplementary material that helps a lot to understand the general picture of a quantum repeater based on this system. The trade-off between entanglement-rate and -fidelity is nicely discussed and the possible steps of improvement included. I recommend its publication.

Version 2:

Reviewer comments:

Reviewer #2

(Remarks to the Author)

The authors satisfactorily answered all my queries. Their explanations are clear and scientifically sound. I am, therefore, glad to recommend the publication of the manuscript.

Reply to the reviewers

We thank all the reviewers for their recommendations, insightful comments, and helpful suggestions to improve our manuscript. We have followed the reviewers' suggestions to revise the manuscript and carefully addressed all the comments raised by the reviewers. In the following, let us address the reviewers' comments point by point:

(Note that in the revised supplementary information with modifications marked, not all the texts of the newly added Supplementary Notes 7-11 and Supplementary Figures 7-14 are marked in red, in order to make them easy to read. We only mark the titles of these Notes and Figures in red to indicate that all the contents in these Notes and Figures are newly added, and leave the other texts still in black. For other parts in the marked supplementary information and the whole marked main text, all the revisions are marked in red.)

Reply to the First Reviewer

Comment: “The manuscript by Zhang and colleagues reports on the generation of heralded atom-photon entanglement based on atomic ensembles used as quantum memories. They use the DLCZ protocol and spatial multiplexing, which is a strong expertise from the authors. The main goal is to enhance the so-called link efficiency, i.e., the ratio between entanglement generation rate and the decoherence time, towards a value closer to a unity. This is a crucial objective for quantum networks, previously demonstrated only in solid-state systems. With atomic ensemble, the achieved values have been one to two orders of magnitude lower. In this paper, the authors play on two parameters to increase this ratio: the number of modes available by combining individual memory cells ($\times 100$) and k-vector multiplexing ($\times 4$). Furthermore, to be as close as possible as real field deployment, they included frequency conversion to telecom wavelength and a 12 km fiber link. The main claim is a significant increase of the link efficiency to 0.46 for such a fiber length.

I appreciated the work and the elegant implementation. The study is also well-characterized, with interesting technical discussions in the SI. However, I do not favor its publication in Nature Communications.”

Reply: We thank the reviewer for thinking that the goal of our work is a crucial objective for quantum networks, as well as the positive assessment that “I appreciated the work and the elegant implementation. The study is also well-characterized, with interesting technical discussions in the SI.”. Below we will address the concerns of the reviewer point by point.

In the first paragraph, the reviewer mentions that “This is a crucial objective for quantum networks, previously demonstrated only in solid-state systems. With atomic ensemble, the achieved values have been one to two orders of magnitude lower.” Here we want to point out that, the solid-state system can only achieve a link efficiency of unity in a lab-scale quantum network (Ref. [13]). For the case of metropolitan-scale fiber, to our knowledge, there is so far no experiment in any physical platform can achieve a higher link efficiency than in our current experiment. Normally, the dominating source for the significantly reduced link efficiency in the metropolitan-scale case is the drop in the attempting rate of heralded entanglement generation,

due to the long waiting time for transmitting the flying qubit and the heralding signal over the long fibers. To solve this issue in the metropolitan case, we need the enhancement via a multiplexed quantum repeater node, which is realized for the first time on an atomic ensemble system in this work.

Comment: “Indeed, this demonstration comes at the cost of very limited cross-correlations, a crucial parameters for further applications. The reported value of 0.47 corresponds to a value of $g_{si}=2.67$. Although this is above the classical bound of 2 (even if, being so close to this value, the full Cauchy Schwartz inequality should be verified), it is well known that such a value is insufficient to enable further applications as this functioning is not enough in the single excitation regime. Most recent networking demonstrations with doped crystals, for instance, were performed in a regime with $g_{si}\gg 10$. Previous DLCZ-based demonstrations have also much higher correlations. A value close to 2 will not enable demonstrations of atom-atom entanglement or even more entanglement swaping due to the large multi-photon components. According the figures, for a g_{si} of about 10, which would be a reasonable for a proof-of-principle demonstration, the probability drops to 0.05. The claim significantly changes in this scenario.”

Reply: We thank the reviewer for raising these important points and the direction for future improvement, which certainly need to be clarified. We agree with the reviewer that for building a practical quantum repeater in the future, both a high link efficiency and a high g_{si} are needed. However, achieving both a high link efficiency and a high g_{si} over a metropolitan-scale fiber network (>10km) is an ultimate objective, which is basically equivalent to the accomplishment of a practical DLCZ quantum repeater, and is beyond the current ability of any group in the world. In the development of a quantum repeater over the past two decades, the progress and advances are always achieved step by step, and many of the results are published in high-profile journals during this process. Thus we can't expect one paper to solve all these important problems. In the development of a quantum repeater (or any other sophisticated technology), when a new breakthrough is achieved, there are always some parameters which are not as good as the current record. However, this does not compromise the significance of that work, as these parameters can always be improved in the future, but the most important thing is that for the first time we can reach this milestone.

In this work, we achieve the multiplexing-enhanced generation of heralded atom-photon quantum correlation over a metropolitan-scale fiber, which is an indispensable ingredient for building a quantum repeater and demonstrated for the first time in atomic ensemble system. Through the significant multiplexing enhancement of more than two orders of magnitude, unprecedented link efficiency and generation rate of atom-photon quantum correlation over a metropolitan-scale fiber are demonstrated, with both of the link efficiency and generation rate being the current record among all physical platforms. Furthermore, if two of such setups are used in the future to generate a heralded atom-atom entanglement, roughly one (0.92) remote entangled pair can be generated within the memory coherence time, which reaches the link efficiency (0.83) required for the task of deterministic entanglement delivery. A link efficiency of unity can also be viewed as a rough threshold for quantum repeater's scale-up, as the connection of two elementary repeater links via entanglement swapping can only be implemented efficiently when the entanglement generation rate is larger than the memory decoherence rate in each elementary link. Thus this work also constitutes one of very first steps towards fulfilling the efficiency

requirement for a practical quantum repeater and reports the highest link efficiency closest to the scale-up threshold so far, as the second reviewer comments: “Their achievement possesses sufficient novelty and will have a strong impact on a broad range of researchers, especially in the fields of quantum networks and quantum internet.” The first reviewer also mentions in the first paragraph of the comments: “This is a crucial objective for quantum networks”.

The reviewer is concerned about whether the lowest $g_{si}=2.67$ demonstrated in our work can yield an atom-atom entanglement of enough fidelity. Thus we simulate the atom-atom entanglement fidelity as a function of g_{si} in several different scenarios, when two of these atom-photon entanglements are used for generating an atom-atom entanglement in the future, as described in the new Supplementary Note 9. As shown in Supplementary Fig. 11d, a g_{si} of 2.67 can yield an atom-atom entangled state with a fidelity of 53%, which is still over the quantum bound 50%, with the multi-excitation error and noise considered. In another case where $g_{si} = 8.7$ and atom-photon link efficiency = 0.06, an atom-atom entanglement fidelity of 83% and a link efficiency = 0.12 can be achieved at the same time. This result will exceed the performances of all reported heralded atom-atom entanglement experiments over metropolitan-scale fiber (>10km) in both entanglement fidelity and link efficiency, as the current state of the art (Ref. [18, 19, 22]) have a fidelity ranging from 56% to 80% and a link efficiency < 0.01.

The reviewer also mentions that in doped crystal or previous DLCZ experiments, a much higher g_{si} can be reached. However, we want to point out that, only having a high g_{si} is not enough, if we want to build a practical quantum repeater of metropolitan scale, not just to demonstrate some particular properties in the lab. To achieve a multi-node practical quantum repeater on metropolitan scale, one must build a functional repeater node with all the required elements equipped (for instance, multiplexing enhancement, frequency conversion, a g_{si} in quantum regime, enough storage time for remote entanglement heralding, on-demand retrieval, etc., which are all demonstrated in this manuscript), to achieve a link efficiency close to or higher than unity and an entanglement fidelity at least 50%. Having all these elements functioning together means much higher difficulties than only optimizing one parameter, as the second reviewer says: “The experimental technologies used in this study, including the multiplexed atomic cloud, read and write sequence, remote heralding system, and frequency conversion system, are highly advanced.”, and the third reviewer says: “Furthermore, the random-access readout possibility at a selectable time is a crucial step and is well demonstrated in the second readout scheme.”. Functional integration of all these requisites needs delicate design and careful trade-off between different elements which have conflicts in the implementation, to guarantee the functioning of the whole system. Therefore, it is reasonable that some parameter is not as high as in some experiments where this parameter is optimized. In the future, we can improve both of the g_{si} and the link efficiency to 50 at the same time (see the new Supplementary Note 7), based on the result of this work.

Here we use an example to explain why only having a high g_{si} is not enough. For a quantum memory with a very high g_{si} (~ 50) but a short lifetime ($\sim 30\mu s$), the g_{si} will quickly decay to a very low value ($g_{si} \approx 1$, corresponding to an atom-atom entanglement fidelity = 0.25) after the necessary storage time ($\sim 120\mu s$ for 12km fiber) for the remote entangling and heralding, which is below the quantum bound and not able to yield an atom-atom entanglement. Therefore, only the g_{si} achieved with an enough storage time can support the implementation of a practical quantum

repeater, as is the case demonstrated in our experiment. Back to our experiment, if we set the storage time to a short value (several tens of microseconds) rather than up to 250us in the experiment, the achieved g_{SI} would also be significantly improved. However, as this long storage time is a must-have in remote heralded entanglement generation, demonstrating this high g_{SI} without necessary storage is not appropriate. Therefore, here we choose to demonstrate the lower g_{SI} with the necessary storage time, following a stringent heralding protocol required in a practical quantum repeater, which is also an important feature of this work.

In addition, although the g_{SI} in our work is not record-high, it is still above the classical bound, which clearly proves the generated correlation is in the quantum regime. Furthermore, with some improvements in the future based on the current work, we can achieve a performance with a $g_{SI} > 50$ and link efficiency > 50 at the same time, to support the ultimate goal of building a practical quantum repeater, which is discussed in Supplementary Note 7. The reviewer also mentions that a Cauchy-Schwartz inequality should be measured when the g_{SI} is close to 2. However, we want to point out that, $g_{SI} > 2$ is widely used as the proof of the nonclassical correlation by many previous works from different groups (for example, Ref. [42], Nat. Phys. 5, 95 (2009), and Phys. Rev. Lett. 101, 120501 (2008)). In these works, the g_{SI} is also close to 2, and the Cauchy-Schwartz inequality is not measured. $g_{SI} > 2$ is used as the only proof to verify the correlation is in quantum regime in these works. Thus here we follow these works to use $g_{SI} > 2$ as the proof of quantum correlation.

Comment: “I have additional comments that should be addressed to clarify the paper and specific claims.

- Figure 1 contains a lot of information, making it difficult to grasp the protocol and the specific techniques employed. The authors should increase the figure size, or separate it in different figures, to improve. Actually I found figure S3 clearer and easier to understand.”

Reply: We thank the reviewer for raising this important point to help us improve the manuscript. Following the reviewer’s suggestion, we have separated the old Figure 1 into two figures (new Fig. 1 and Fig. 2) in the revised manuscript, to illustrate the protocol and multiplexing techniques in a clearer way.

Comment: “-The retrieval efficiency is limited to the percent level. This is very low and not compatible with quantum networks where photons need to be read out and synchronize with others. This is a strong limitation for the scalability of the scheme. The authors should address this point in greater detail (no values of optical depth are given for instance). Similarly, in the conclusion, the potential increase of the memory lifetime via the use of optical lattices is discussed. What about the achievable efficiency and vector multiplexing in that case as the memory cell will have a much reduced extension ?”

Reply: We thank the reviewer for raising these important points. First, we want to clarify that the retrieval efficiency we demonstrate in the old Fig. 1d (Fig. 2d in revised manuscript) is the overall retrieval efficiency which includes all the losses and inefficiencies in the atom, optical elements, and detectors. If we only consider the intrinsic efficiency with a very short storage time, the intrinsic retrieval efficiency on the atom is about 15%, which is slightly lower but still reasonable if compared to the typical value of 20-30% in free-space atomic ensemble system. We

attribute the slightly lower intrinsic efficiency to the optical depth drop due to imperfect optical pumping, and the atom loss also in the optical pumping process. The typical optical depth is about 5 on the center cells and about 3 on the edge cells in the memory array, and this relatively low optical depth is the major limit for the intrinsic retrieval efficiency. In the future, it is possible to achieve a high intrinsic efficiency at the level of 50% in each memory cell, by loading the atoms into an optical lattice array (thus much higher optical depth), and employing wavepacket shaping techniques during the read-out. We have added the information about optical depth and optical pumping into Supplementary Note 1, and an analysis of how to improve the retrieval efficiency in the future into Supplementary Note 7.

Another major source for the relatively low overall efficiency is the inefficiencies of the optical elements used in this experiment, such as the two-dimensional AODs for individual addressing, the EOMs for on-demand retrieval of one of the four angular modes, additional filter etalons due to the small angle between the idler photon mode and read laser for long memory lifetime, a couple of fiber connections in the middle just for convenience, and the low efficiency of the detector. These inefficiencies contribute to a low total efficiency of roughly 17%, and there's a great room for future improvement. We list all the efficiencies in the full optical path in detail in Supplementary Fig. 1f. We also discuss how to improve the transmission in the optical path in the future in Supplementary Note 7, and we can see that it's possible to achieve a high overall retrieval efficiency at the level of 25% in the future.

In addition, it is also worth to note that, for the heralding and swapping scheme based on the single photon detection in the DLCZ quantum repeater protocol, the influence of the inefficiency in the read out is much less significant compared to those protocols where entanglement swapping is based on two photon coincidences. Besides, for a future large-scale practical quantum repeater, the retrieval efficiency only appears as a constant coefficient in the final polynomial for the communication time (Ref. [2]), which does not influence the fundamental achievement for a quantum repeater which accelerates the remote entanglement distribution from exponential time cost to polynomial cost. Nevertheless, we still agree with the reviewer that it is important to improve the retrieval efficiency in the future, and a discussion of achieving a high overall efficiency of 25% is added into Supplementary Note 7.

Regarding the achievable efficiency and vector multiplexing when a lattice array is applied in the future, first we want to point out that the current optical configuration does not have to be changed if optical lattice is employed. In previous works where 1D optical lattice is employed (for example, Phys. Rev. A 81, 041805 (2010)), the size of the atomic cloud loaded in the lattice is quite large (~130um in radius), with a signal/idler mode of 110um (Gaussian radius) used in that work, which is already larger than the size of signal/idler mode which is 60um in our current work. Thus we don't have to change the beam configuration when the optical lattice array is applied. In Supplementary Note 7, we estimate with a moderate optical depth of 30 in each optical lattice, together with wavepacket shaping, a high intrinsic retrieval efficiency of 50% can be achieved. As the beam configuration is not changed, the same vector multiplexing can also be achieved with optical lattice array applied.

Comment: “-One of the technique here to increase further the multiplexing is to use k-vector multiplexing. I m curious about the effect of this technique on added noise.”

Reply: We thank the reviewer for raising this important point. k-vector multiplexing is a mature multiplexing method, which have been widely used by many groups in the world (such as Nature 454, 1098 (2008), Phys. Rev. Lett. 119, 130505 (2017), Ref. [42]), and also by our group (Ref. [26]) previously. Regarding the added noise, the crosstalk between each mode will be negligible if the angular separation between them is larger than the divergence angle of each mode (as each mode is a Gaussian beam). In our current experiment, this requirement is fulfilled as shown in the inset at the bottom left of Supplementary Fig. 1a, that at the idler AOD (where the angular distribution of each mode at the atoms is mapped to spatial distribution), the four k-vector idler modes can be clearly separated, with an estimated crosstalk to each other less than 1%. We also confirm the crosstalk by measuring the cross correlation function between signal and idler photon modes between these four different k-vector modes, via the write and read in DLCZ protocol. As shown in Supplementary Fig. 1c, the cross correlation between different k-vector modes are all around 1, and is larger than 10 if the signal and idler photon is from the same mode, which provides the evidence that the crosstalk is negligible. Except the potential crosstalk, we haven't observed any other added noise.

Comment: “-The authors increase the storage lifetime by optical pumping, further described in the SI. What is the efficiency of the process that involves microwave pulses and the final optical depth to be used for the DLCZ scheme?”

Reply: We thank the reviewer for raising this important point. The overall efficiency of preparing the atoms to the $|F=2, mF=0\rangle$ state is about 88%, including the fidelity of the initialization to the state $|F=1, mF=1\rangle$ of 93%, and the fidelity of the microwave pi pulse of about 95%. After optical pumping, the optical depth is about 5 in the center cell and about 3 in the edge cell. We have added these parameters into Supplementary Note 1.

Reply to the Second Reviewer

Comment: “In this article, the authors demonstrated the non-classicality of the atom-photon cross-correlation function over a 12 km optical fiber with a quantum link efficiency of 0.46. By integrating single-photon interference using current technologies in the future, this value can be doubled, surpassing the value of 0.83 required for deterministic delivery of quantum entanglement.

To enhance the quantum link efficiency, improving the entanglement sharing rate is crucial. To increase the number of trials, the authors developed a 280-mode multiplexing of the atomic cloud by combining 70 spatial modes, each with 4 angular modes. Additionally, they used quantum frequency conversion to convert 795 nm signal photons emitted from the atomic cloud into telecom photons, thereby improving the sharing rate per mode.

There is a trade-off between the success probability and the value of the cross-correlation function. By quantitatively analyzing this relationship and setting the appropriate probability, they achieved a link efficiency of 0.46 with a g_2 value of 2.67 for a fixed storage time of 130 μ s. They also performed another protocol with a user-defined retrieval time ranging from 130 μ s to 250 μ s. In this case, the link efficiency was 0.30, slightly lower than 0.46, but they discussed potential improvements, such as enhancing the frequency conversion efficiency and introducing

an optical lattice array.

Their results highlight the advantages of quantum repeaters over long distances. The experimental technologies used in this study, including the multiplexed atomic cloud, read and write sequence, remote heralding system, and frequency conversion system, are highly advanced. The experiments and results are clearly presented and easy to follow for readers including the Supplemental material. Their achievement possesses sufficient novelty and will have a strong impact on a broad range of researchers, especially in the fields of quantum networks and quantum internet.

As described below, I have questions about the quantum link efficiency which is one of the most important results. Including this, I listed several comments and questions below. After they properly address them, I recommend the publication of the manuscript in Nature Communications.”

Reply: We thank the reviewer for the very positive comments, strong recommendation for publication, and for raising important points that help us to clarify and improve our manuscript. In the following, we address the reviewer’s comments/questions point by point:

Comment: “Major comments:

Q1

The cross correlation of 2.67 was observed in the protocol of the fixed storage time. How much value of the fidelity to the entanglement after the entanglement swapping based on single-photon interference is expected from the observed value?”

Reply: We thank the reviewer for raising this important point. We have added a simulation of the heralded atom-atom entanglement fidelity as a functional of the cross correlation under several different schemes into Supplementary Note 9 in the revised Supplementary Information. As illustrated in Supplementary Fig. 11d, the atom-atom entanglement fidelity after the single-photon interference is 53%, with $g_{si}=2.67$ for each of the two atom-photon correlations, which is still higher than the quantum threshold of 50%. We have also listed the corresponding atom-atom entanglement fidelities of the measured data points with different atom-photon cross correlation in the new Fig. 4d and Fig. 5d.

Comment: “Q2

Related to Q1, quantum link efficiency in Eq.(1) does not appear to include the amount of entanglement. This may allow for a strategy where generating low-quality entanglement with high efficiency becomes advantageous. However, in Eqs.(1),(2) and Fig.1b of Ref.[13], the quantum link efficiency should be related to the average fidelity with respect to the maximally entangled state. It is natural to consider the cost of entanglement distillation. This paper seems to lack the consideration to fill this gap. The authors should clearly explain the point”

Reply: We thank the reviewer for raising this insightful point. We agree with the reviewer that for a practical quantum repeater, both the efficiency (link efficiency) and the quality (fidelity) are crucial parameters, which should be taken into consideration simultaneously. Similar to Fig. 1b

of Ref. [13], we have added a simulation which illustrates the achievable atom-atom entanglement fidelity as a function of the cumulative success probability in the new Fig. 6 of the revised manuscript, under several different memory performances. The trade-off between the efficiency and the quality is clearly illustrated in Fig. 6, and a new paragraph before the discussion part in the main text is also added to demonstrate the trade-off.

In addition, we have also analyzed the performance of entanglement distillation in a case where multimode quantum repeater nodes with high link efficiency are employed to distill the generated remote quantum entanglement, as described in Supplementary Note 10 and Supplementary Fig. 13. By this scheme, multiple pairs of remote entanglements can be generated within memory coherence time thanks to the high link efficiency and the large capacity of the multi-mode quantum memory, and the fidelity of the remote entanglement can be improved by entanglement distillation.

Comment: “Other comments:

Q3

In Eq.(1), the enhancement factor of the success probability related to T_{ent} is considered. Is it possible to consider the enhancement of the memory time T_{coh} , for example, by including the swapping operation of the quantum state stored in the memory to fresh memory?”

Reply: We thank the reviewer for raising this insightful point. As the reviewer points out, in a fully functional multiplexed quantum repeater in the future, one can continuously prepare many copies of remote entangled memory pair and store them for further use, which can effectively prolong the coherence time of the memory due to this redundancy of entanglement resources. For example, by replacing the significantly degraded entangled pair generated in the early time with the most recently generated entangled memory pair stored in the multi-mode memories, an entanglement with higher fidelity can be delivered, which is equivalent to prolonging the memory coherence time. As the reviewer points out, this refreshing scheme can be viewed as another interpretation of the multiplexing enhancement to the link efficiency in Eq. (1), that the memory coherence time T_{coh} is effectively prolonged by N times, which can yield the same result via our current interpretation of reducing T_{ent} by N times. We have added this alternative interpretation into the introduction part.

Furthermore, in Supplementary Note 10 and Supplementary Fig. 13 we have also analyzed the above scheme in detail. In this scheme, the entanglement fidelity is improved by replacing the entanglement generated in an earlier time with the newly generated one, which clearly demonstrates the advantage of a multi-mode quantum repeater over a single-mode quantum repeater.

In addition, a detailed analysis of how multiplexed quantum memories can improve the performance of a quantum repeater is demonstrated in Ref. [31].

Comment: “Q4

The link efficiency for the deterministic delivery of entanglement is 0.83, which is lower than 1. Could the authors provide an intuitive explanation for why a value less than one is sufficient?

Such a description would be helpful for readers.”

Reply: We thank the reviewer for raising this important point. As the reviewer points out, intuitively, the threshold for deterministic delivery of entanglement should be around link efficiency = 1, as the entanglement generation rate is same as the entanglement decoherence rate in this case. In the following intuitive but very rough estimation, we consider the case where the link efficiency = 1. Here we set the pre-defined delivery time to the memory coherence time T_{coh} (also equals to T_{ent} since link efficiency = 1). In this situation, we would have a high probability to generate an entangled state (as the protocol has a length of T_{ent}), and the generated entangled state also suffers from remarkable decoherence since the storage time is on the scale of T_{coh} on average. Therefore, we would expect that the final output is an entangled state with not so much amount of entanglement remained, which corresponds to an average fidelity of slightly higher than 0.5 (the threshold of quantum entanglement). Therefore, we would expect that link efficiency = 1 can roughly yield an average fidelity slightly higher than 0.5, thus a slightly lower link efficiency can roughly guarantee an average fidelity of 0.5. Note that in this rough estimation we have ignored the chance of unsuccessful entanglement generation. We have added this intuitive explanation into the introduction part.

Comment: “Q5

How are the details of the etalons used for the detection of idler photons and for frequency conversion? How are they stabilized?”

Reply: We thank the reviewer for raising this important point. The etalons for the detection of idler photons are customized fused silica solid etalons (Foctek Photonics), with a free spectral range of 13.6GHz, and a finesse of 30. The transmission efficiency for the signal/idler photon is about 90%, and the elimination is roughly 27dB for the write/read laser 6.8GHz away. For frequency conversion, we use another customized air-spaced etalon (SLS optics) with ULE substrates, with a free spectral range of 10GHz and a finesse of 150. The transmission efficiency is about 72%. All the etalons are in a plano-plano design. All the etalons in this experiment are actively stabilized by controlling the temperature of the homemade aluminum enclosure of the etalon, with a precision better than 0.01K. We have added these information regarding the etalon into Supplementary Note 1.

Comment: “Q6

In Fig. 1d, the cross-correlations of the signal and idler photons were around 10. These values seem relatively small compared with conventional experiments based on atomic clouds. What is the main cause of the degradation of the cross-correlation functions?”

Reply: We thank the reviewer for raising this important point. The first source for this relatively small cross correlation is the background noise induced by the pumping laser in the frequency conversion, which is a constant value at about 200Hz. If the signal photon excitation probability is too low, the signal-to-noise ratio of the signal photon after frequency conversion would also be very low. Thus to achieve a good signal-to-noise ratio, we need to use a relatively high excitation probability, which induces the relatively low cross correlation. We have added an analysis about the background noise of the frequency conversion, as well as the method to reduce the

background in the future in Supplementary Note 7. In the analysis we show that, by slightly changing the wavelength of the pumping laser by 6nm, we can achieve a background noise which is 1/5 of the current value, and thus 5-fold improvement in the signal-to-noise ratio, with the conversion efficiency still the same. We can also improve the signal-to-noise ratio by improving the total efficiency of the signal channel in the future. With the improvement in the signal-to-noise ratio, we can reduce the excitation probability and achieve a high cross correlation.

Another source for the small cross correlation is the relatively low intrinsic retrieval efficiency ~15% mainly due to the low optical depth in our experiment. This problem can be solved by loading the atomic cloud into optical lattice array in the future, as also discussed in Supplementary Note 7. We estimate an intrinsic retrieval efficiency of 50% can be achieved with a higher optical depth of 30 due to optical lattice array. With all these improvements, we estimate a high cross correlation over 50 and a high signal-to-noise ratio over 50 can be achieved at the same time in the future.

Comment: “For Supplemental material:

Q7

On L9 and L19, references should be added to “in our works” and “in our prior work.””

Reply: We thank the reviewer for pointing out this important point. We have added the related references to these places in the revised Supplementary Information.

Comment: “Q8

On L22, the authors explain the angle between the write beam and the signal photon is 0.3 degrees. Is this small angle resolved by increasing the path lengths?

If so, could the authors tell me the lengths of the photons and the write beam? If different techniques are used, this should be clarified.”

Reply: We thank the reviewer for raising this important point. In this work we do not use the angular separation to resolve the write/read laser from the signal/idler photon, as this angle separation of 0.3 degrees is at the same level of the divergence angle of the signal/idler mode (the Gaussian radius is 60um at the waist, thus a half divergence angle of roughly 0.25 degrees). Thus there is always substantial overlap between the write mode and the signal modes, no matter how long the path is. Thus we cannot resolve the write and signal via increasing the path length. In the experiment we combine and split the write/read laser with signal/idler mode by two 9:1 beamsplitters (with some sacrifice in efficiency), as illustrated in Supplementary Fig. 1a. In addition, it is not feasible to filter out the write/read laser from the signal/idler photon by angular separation in photon detection. Instead, we filter the unwanted write/read laser by spectrum method via three cascaded filter etalons. Each of the etalon can provide an extinction ratio of about 27dB, thus 80dB of extinction of the noise laser can be achieved. Together with another 20dB from the polarization separation and another 10dB from the angle (still 0.3 degrees separated), we can achieve a total extinction ratio of 110dB to the write/read laser at the end, which is enough to eliminate the noise from write/read laser in both signal or idler photon. We have added these discussions in Supplementary Note 1.

Reply to the Third Reviewer

Comment: “In this manuscript a significant technological difficulty is addressed to establish the required entanglement rate for a fully operational quantum repeater. The quantum link efficiency in the continuous picture of [1] is used as figure of merit and it is shown that the experimental realisation is close to the required value to become fully operational in terms of decoherence and ‘quantum correlation’ rate. Especially it is shown that two such devices will beat the required value of 0.83, which is the benchmark to get in the mean one entangled state with a Fidelity >0.5 per attempt. The protocol is the one photon DLCZ protocol of [2], and the experimental key element to create the necessary rate is by applying on demand addressable spatial and angular multiplexing methods, all coupled to a single mode fibre and frequency converted to the telecom C-band.

The results are convincing and the figures appropriate to understand the experiment. It is remarkable that with this experiment one frequency band of a 12km long transmission fibre is temporally filled and it seems that it is possible to go beyond by a factor 100/70 by using all the 100 memory cells instead of only 70. Furthermore, the random-access readout possibility at a selectable time is a crucial step and is well demonstrated in the second readout scheme.”

Reply: We thank the reviewer for the very positive comments, and for raising important points that help us to clarify and improve our manuscript. In the following, we address the reviewer’s comments/questions point by point:

Comment: “My main criticism in this manuscript is a missing discussion regarding the entanglement generation rate and fidelity to address the full picture of an operational quantum repeater, especially as the realization becomes closer. To obtain entanglement with this protocol, it is necessary to combine two such devices -which they call segment- to an entangled memory-memory segment in an interferometric way. Certainly, difficulties arise which should be discussed. In this regard they point out issues with another system (line 69-74: ‘Meanwhile, issues like unwanted spontaneous emission and cavity jitter limit the indistinguishability of the heralding photon [21, 36], which need to be suppressed in the future to guarantee high-quality ion-ion entanglement without harming the efficiency significantly [21].’) but a discussion to convert the described ‘quantum correlation’ into the memory-memory entanglement is missing and should be added. I see difficulties in the required interferometric stability, starting with the two excitation lasers until the overlapping of the photons, as there are many different paths involved and also the frequency conversion is included. Also, the frequency shift due to the deflectors should be discussed in this sense. Maybe the discussion should go beyond, if two memory-memory segments get combined in the random-access manner that is used in the second readout scheme. It would be also interesting if all memories can be connected, i.e. if the spatial multiplexing allows the entanglement swapping between any of the memories.”

Reply: We thank the reviewer for raising these important points. We agree with the reviewer that both of the entanglement generation rate and the fidelity are crucial for an operational quantum repeater and should be discussed. In the revised manuscript, we have added a detailed analysis of how to combine two of such atom-photon quantum correlations into an atom-atom entanglement via several different schemes in Supplementary Note 9 and Supplementary Fig. 11 of the revised

supplementary information, as well as the entanglement connection of two memory-memory segments to implement a three-node quantum repeater (Supplementary Note 9 and Supplementary Fig. 12). In addition, the trade-off between efficiency and fidelity under different conditions are also analyzed (see Fig. 6 in the revised manuscript).

Furthermore, we agree with the reviewer that the difficulties in generating an atom-atom entanglement should be discussed, as in Ref. [21]. Thus we have also carefully analyzed the technical difficulties such as the required interferometric stability in single photon interference (including the frequency conversion), with the different paths, the frequency shift due to AOD addressing, and photon overlapping taken into consideration, in the new Supplementary Note 8.

As the reviewer suggests, we have discussed the performances of different swapping schemes ('parallel case' and 'multiplexed case') between two memory-memory entanglements in a 3-node multiplexed quantum repeater in Supplementary Note 8. It is also worth to note that the performance of these schemes have been carefully analyzed in Ref. [31].

Comment: "Meanwhile a new result appeared (arXiv:2406.09480v1), it may be interesting how this compares."

Reply: We thank the reviewer for bringing this new result to our attention. This paper demonstrates a 10-ion node and the multiplexing capability through ion shuttling. The ten ions are excited one by one to generate a 10-mode time-bin train into a single mode fiber, which is very similar to our current work. The fidelity of each ion-photon entanglement is measured, and the efficiency inhomogeneity of each mode is carefully analyzed. Although some required elements for multiplexing-enhanced heralded ion-photon entanglement over a long fiber which are demonstrated in our work are not shown (for example, frequency conversion and long fiber), this work still constitutes a significant advance in the development of trapped-ion quantum network, and represents the state of the art in the implementation of a multiplexed trapped-ion quantum network node. In addition, we think the combination of cavity and multiplexing enhancement demonstrated in this work is a very promising approach towards the future functional quantum network, and the shuttling scheme used in this experiment is elegant and novel. This work is also an evidence that the multiplexing enhancement scheme used in our work is widely regarded as a key approach to improve the performance of a quantum network node, and shares broad interest among research groups working on different physical platforms, not only limited in atomic ensemble system.

We have added this important work as Ref. [35] in the reference list of the revised manuscript.

Comment: "Further questions and suggestions:

Line 39-42:

'It is also noteworthy that η_{link} is twice the expected number of heralded atom-photon correlation generated within memory coherence time over L , if Type I protocol (single photon interference) is used [2, 3, 13, 15, 18, 24, 30] (by receiving clicks from both detectors at the detection station, see Fig. 1a).'

This sentence causes a few problems to me:

a) I don't understand the sentence: 'eta-link is twice the expected number of heralded atom-photon correlation generated within memory coherence time over L'

b) The terminology 'Type I protocol' is -in my opinion- not clear and can be avoided as it is not used in the rest of the manuscript. A short reminder on formula (1) of [2] and the link of $\sqrt{p_c}$ to the success probability (\bar{p}) of this manuscript would help to understand the rest of the manuscript much faster.

c) '(by receiving clicks from both detectors at the detection station, see Fig. 1a).' The detector station is in Fig. 1.b and there is only one detector in the experiment."

Reply: We thank the reviewer for raising these important points. For comment a), here we are describing the fact that, the link efficiency of generating a heralded atom-atom entanglement by combining two atom-photon quantum correlations through single photon interference should be $2\eta_{\text{link}}$, if the link efficiency of each heralded atom-photon correlation is η_{link} (see Supplementary Note 11 for details). The reason behind is that, the heralding (success) probability doubles but the time cost is still the same in each attempt, when we combine two atom-photon correlations into an atom-atom entanglement. The 'expected number of heralded atom-photon correlation generated within memory coherence time' is another expression of link efficiency η_{link} in the heralded atom-photon correlation generation, as explained in Supplementary Note 11.

We have added a detailed analysis of connecting two atom-photon correlations into an atom-atom entanglement via single photon interference, in Supplementary Note 11. We analyze the resulted atom-atom entangled state heralded by click on each detector, the success probability of each detector click, and the total chance of a successful heralding with two detectors. We have referred to Supplementary Note 11 in the main text.

For comment b), we thank the reviewer for this suggestion to improve the readability of our manuscript. We have removed the phrase 'Type I protocol' in the manuscript as the reviewer suggested. We have also revised the introduction part to link the success probability of this manuscript to the success probability of formula (1) of the original DLCZ paper (Ref. [2]).

For comment c), we have revised Fig. 1a to highlight the presence of the middle detection station. There are two detectors receiving the photons from both output channels of the beamsplitter, and the click on either detector can herald an atom-atom entanglement. Here in this experiment, we only implement the atom-photon quantum correlation between a quantum repeater node and a signal photon on one side in an elementary link, without the interference of the photons from both sides on the beamsplitter in the detection station for atom-atom entanglement. To measure the quantum correlation of one side, the beamsplitter is not needed. Thus we can replace the beamsplitter and two detectors with one detector as depicted in Fig. 1b and Supplementary Fig. 14. We have revised the caption of Fig. 1 to clarify this point. We have also added a detailed description and analysis in Supplementary Note 11 and Supplementary Fig. 14.

Comment: "Line 85-88

'This corresponds to a $\eta_{\text{link}}=0.92$ if two such heralded atom-photon quantum correlations are

employed to generate a 24 km atom-atom entanglement via single photon interference [2, 3, 13, 15, 18, 24, 30] in the future, which will be close to the scale-up requirement to build a multi-node quantum repeater.’

I don’t understand why eta doubles when the link gets extended at the same time. In my opinion this is only the case when two such devices are used to bridge the same distance. Can you explain that?”

Reply: We thank the reviewer for raising this important point. In our definition, the $L=12\text{km}$ is the distance between one memory and the detection station, which is located at the middle point of an elementary link with a length of $2L=24\text{km}$. Thus here 12km is a half of an elementary atom-atom entanglement of 24km . If a pair of such 12km atom-photon correlation is combined to generate an elementary atom-atom entanglement as shown in Supplementary Fig. 14, the distance from memory A to the middle detection station is $L=12\text{km}$, and the distance from middle detection station to memory B is also $L=12\text{km}$. Thus the distance from memory A to memory B is $12+12=24\text{km}$, if we assume memory A, the middle detection station, and memory B are on the same line. At the same time, the success probability of heralding an 24km atom-atom entanglement is twice the success probability of a single atom-photon correlation of 12km , as described in reply of last comment. Therefore, when we combine two atom-photon quantum correlations into an atom-atom entanglement, both the link efficiency and the distance are doubled.

We have revised Fig. 1a and added a new Supplementary Fig. 14 and Supplementary Note 11 in the supplementary information to make this point more clear. We have also referred to Supplementary Note 11 in the main text.

Comment: “Line 91

‘... equals to 1.95 kHz during each 240 us protocol ...’.

This formulation is not clear, I guess you mean when excluding MOT loading and pumping?”

Reply: We thank the reviewer for raising this important point. The reviewer is correct that this 240us protocol only include the 120us excitation stage for the successive excitation of the 280 memory modes and the 120us heralding stage for receiving the heralding signal of each mode, as shown in the new Fig. 4a. This 240us can be regarded as a single complete protocol for the excitation and heralding of the atom-photon quantum correlation, with all other time costs such as the MOT loading and pumping viewed as the overhead. We have emphasized that the MOT loading and pumping is excluded in the introduction part.

Comment: “Line 112

‘Here each memory cell is a micro-ensemble of Rb87 atoms, ...’

It would help if you specify the trap, i.e. the realisation of the micro ensembles (‘... different memory cells are different parts of a single atomic cloud ...’ taken from [38], is a tweezer array used?)”

Reply: We thank the reviewer for raising this important point. In our experiment, different

memory cells are different parts of a single free-expanding atomic cloud, and these 100 cells are not loaded into optical trap arrays, which is different from the tweezer array experiments. We have revised the corresponding part of the manuscript to emphasize this point to avoid misunderstanding.

Comment: “First paragraph of results (line 108-130)

Refer to Fig 3a and describe the time overhead of the protocol (MOT loading and optical pumping).”

Reply: We thank the reviewer for raising this important point. We have referred to the new Fig. 4a and described the time overhead of the protocol here.

Comment: “line 141

I would suggest to mention the frequency shift and the memory-cell selection due to the AO-deflectors. Especially if the interference is affected.”

Reply: We thank the reviewer for raising this important point. We have added a discussion there to describe the frequency shift and memory-cell selection due to the AOD, as well as the influence to the single photon interference for atom-atom entanglement in future.

Comment: “Line 148

How is the SNR defined? And wouldn't it be better to specify the noise photons in dependence of the pump power?”

Reply: We thank the reviewer for raising this important point. Here in this work, the signal is the detector clicks induced by signal photon in the detection window, and the noise includes the detector clicks in the detection window induced by the noise photon in the frequency conversion stage (~200Hz due to strong pumping laser), the dark count of the SNSPD (~10Hz), and all other sources of noise photon (e.g. room light leakage). Both of the signal and noise are measured in the final SNSPD after the frequency conversion. We have replaced the graph of SNR in Fig. 2e with the noise versus power, as suggested by the reviewer in the later comments. At the same time, the SNR figure is moved to Supplementary Fig. 7b. A detailed setup for frequency conversion and the description of the SNR versus pumping power is added in Supplementary Note 7.

Comment: “Line 165-170

‘This protocol demonstrated here simulates a real-world ... via single photon interference.’

Should be discussed -and further explained- in the introduction part.”

Reply: We thank the reviewer for raising this important point. We have added several sentences here and in the last paragraph of introduction to discuss and referred to Supplementary Note 8-11 to explain how to implement heralded atom-atom entanglement based on two of such setups.

Comment: “Line 187-189

‘After the successful heralding in any of the 280 modes, we read out the stored spin-wave mode for further applications. Here we demonstrate the retrieval of the quantum information stored in the quantum memory’

Change to: ‘After the successful heralding in any of the 280 modes, we read out the stored spin-wave mode to demonstrate the retrieval of the quantum information stored in the quantum memory’”

Reply: We thank the reviewer for raising this important point to improve the readability of our manuscript. We have implemented this change suggested by the reviewer in the revised manuscript.

Comment: “Line 198

‘In the first read-out style, the storage time is fixed, but the read-out time is random.’

It is always a programmed time, so I don’t understand which time is random.”

Reply: We thank the reviewer for raising this important point to improve our manuscript. In the first read-out style, the storage time, which is the interval between the read-out time and the excitation time, is a constant (round-trip travel time for signal photon and heralding TTL+ time for identifying the excited mode and read out = $120\mu\text{s} + 10\mu\text{s} = 130\mu\text{s}$) for each of the 280 modes. Here the random read-out time means the time t when the read-out operation is executed. Here we define the beginning of the excitation stage in the protocol as $t = 0$, as shown in Fig. 4a. As we successively excite each spatial mode into a time-bin train, the excitation time for different mode is different, thus the read-out time for different mode is also different. For example, the excitation time for the 1st mode is $t = 0\mu\text{s}$, and the excitation for the 280th mode is at $t = 119\mu\text{s}$ in the excitation stage. Then the read-out time for the 1st mode is $t = 0 + 130 = 130\mu\text{s}$, and the read-out time for the 280th mode is $t = 119 + 130 = 249\mu\text{s}$, which are different in time t but have the same storage time $130\mu\text{s}$. We have revised the corresponding parts to explain this in a clearer way.

Comment: “Line 210-212

‘Here we choose to read out at most 3 excited spin wave modes during each heralding stage for accelerated data collection.’

Why is the data collection accelerated?”

Reply: We thank the reviewer for raising this important point to improve our manuscript. The data collection means the record of the signal-idler photon coincidence, thus we need to collect as many signal-idler photon coincidence in each $240\mu\text{s}$ protocol as possible. In a $240\mu\text{s}$ protocol, sometimes more than one signal photons can be recorded from different memory modes, thus we can read out all these excited memory modes into idler photons to accelerate this process. For example, if the excitation in two memory modes are herald by recording corresponding signal photons during a single heralding stage, we can read out both of them to idler photons one by one. By this scheme we can collect signal-idler coincidences two times faster in this $240\mu\text{s}$ protocol of excitation and heralding, compared to the scheme of only reading one of them (and

discard the other excited mode). In both Fig. 4a and Fig. 5a, we demonstrate the read out of two modes in a single 240us run. In the situation that the p_{total} is large (for example, $p_{\text{total}} = 0.47$), the chance of heralding more than one modes is not negligible. Thus by this scheme we can accelerate the data collection rate.

Another consideration is that, creating and storing many copies of entanglement between many pairs of memory modes is an advantage of a multimode quantum repeater. To exploit this advantage, the ability of reading out many excited memory modes in a programmed way is necessary, and is demonstrated in this way. Here the different quantum correlations between different pairs of signal photon and memory modes are independent to each other, all of these can be used for further applications. When the excitation probability gets high, for example, for link efficiency close to unity, the chance of creating more than one excitation is high. Since all of these generated correlations are useful in later application, it is natural all of them should be measured, to demonstrate the ability to individually control each excited memory mode, and to accelerate the data collection at the same time.

We have added the explanation for the acceleration in data collection into Supplementary Note 10. We also refer to Supplementary Note 10 in the main text.

Comment: “Line 225

‘... varying average success probability ...’

how is the probability set and which excitation probability does this correspond (and which fidelity)?”

Reply: We thank the reviewer for raising this important point. The intrinsic excitation probability is set by adjusting the power of the write pulse to excite each memory mode, and the excitation probability is roughly linear with the write power.

The success probability per mode (x axis) of the data points demonstrated in Fig. 4b and Fig. 5b is the overall success probability (the probability of recording a detector click every excitation attempt on a memory cell), which includes the contribution from the intrinsic excitation probability and the efficiency in the whole optical path (optical elements, frequency conversion, and detector). We have also done a thorough analysis of the correspondence between g_{SI} and the entanglement fidelity, in Supplementary Note 9. We have listed the corresponding intrinsic excitation probability and atom-atom entanglement fidelity of each data point in Fig. 4b, (Fig. 5b), into a list in the new Fig. 4d, (Fig. 5d). As the data points in Fig. 4c (Fig. 5c) have similar success probabilities, they are not listed to avoid repeating.

Comment: “Line 243

Refer to the measurement in the supplementary material for the coherence time as this is important for η_{link} . I would suggest explaining in the supplement how this is measured.”

Reply: We thank the reviewer for raising this important point. We have referred this place to the measurement in the supplementary information. We have given a detailed description of how to measure the coherence time in the revised Supplementary Note 1.

Comment: “Line 245-249

‘... which also corresponds to an equivalent quantum link efficiency $\eta_{\text{link}} = 0.46 \times 2 = 0.92$ if two such setups are combined to generate heralded atom entanglement with single photon interference in the future.’

See comment to line 85-88”

Reply: We thank the reviewer for raising this important point. We have explained this in the reply to comment regarding line 85-88.

Comment: “Line 258

‘the memory always reads out the stored spin wave mode at a user-defined timestamp’

Check formulation and the term ‘timestamp’ is difficult to understand.”

Reply: We thank the reviewer for raising this important point. This sentence means no matter which memory mode is excited, it can be read out at an arbitrary time t ($t = 250\mu\text{s}$ for example) defined by the user. We have revised the corresponding place to clarify this point.

Comment: “Line 261-263

‘Note that in this protocol the read-out time is fixed no matter which mode is excited, meanwhile the storage time is variable (for example, the storage time is 250 μs for the 1st mode, and 130 μs for the 280th mode)’

Suggestion: ‘As a consequence, the storage time is variable for example, 250 μs for the 1st mode, and 130 μs for the 280th mode.’”

Reply: We thank the reviewer for raising this important point to improve our manuscript. We have implemented this revision as the reviewer suggested.

Comment: “Line 272

‘... for faster data collection.’

See comment to line 210-212”

Reply: We thank the reviewer for raising this important point. We have explained this in the reply to comment regarding line 210-212.

Comment: “FIG. 1

- a) In Fig 1b: typo: ‘E/O convertor’
- b) In Fig 1d: Can you add dimensions to the array?
- c) In Fig 1d: The label of the cross correlation graph is not explained.
- d) In Fig 1d: Instead of SNR, I would suggest showing the noise photons and add the

corresponding SNR to the p values in the text.

e) Label: ‘This experiment represents half of the heralded entanglement generation ...’ In my opinion, there are components missing for the necessary phase stability, in this sense it is not half of the necessary components. I suggest finding another formulation for your experiment.”

Reply: We thank the reviewer for raising these important points to improve our manuscript.

- a) We have corrected this typo.
- b) We have added x, y, z axes and the corresponding AOD frequencies to the array. We have split previous Fig. 1 into two figures as suggested by the first reviewer. Currently this figure is in Fig. 2b in the revised manuscript. We also put the corresponding Cartesian coordinate in the new Fig. 2a.
- c) We have added the explanation for the label of the cross correlation in Fig. 1d (new Fig. 2c). The orange circle represents the memory cell (with $f_x = 103\text{MHz}$, and $f_y = 103\text{MHz}$) in the center region of the array, and the emission angle of the signal photon is in the A direction. The blue triangle and the purple diamond are also described in the caption.
- d) We have revised the new Fig. 2e to show the noise photons and moved the SNR graph to Supplementary Fig. 7b, and revised the corresponding caption.
- e) We have changed this sentence to ‘This experiment represents part of the heralded entanglement generation in a long elementary link of a multiplexed quantum repeater. With two such remote atom-photon quantum correlations and the phase stability techniques, deployed fiber, and the synchronization of control systems demonstrated in recent works [18-23, 36], the heralded atom-atom entanglement over a metropolitan-scale elementary link can be established via single photon interference in the future.’ in the caption of Fig. 1a.

Comment: “**FIG. 2**

- a) Specify the bin size of the histogram.
- b) ‘Note that the excitation probability here is higher than in Fig. 1e, thus higher signal-to-noise ratio is achieved.’ This sentence can be omitted if the noise photons are specified.”

Reply: We thank the reviewer for raising these important points.

- a) The bin size of the histogram is 10ns. We have added this information into the caption.
- b) Since we have specified the noise photons in the new Fig. 2e, this sentence has been deleted.

Comment: “**Supplementary material:**

Line 9

‘... previous works.’ : add references”

Reply: We thank the reviewer for raising this important point. We have added references to this place.

Comment: “In line 52

What do you understand as the feedforward process? This name confuses me, as the readout is always performed after herald detection and not ‘foreseen’ from prior runs.”

Reply: We thank the reviewer for raising this important point. We apologize for misusing the word ‘feedforward’ here. We have changed the ‘feedforward process’ to ‘process for mode identification and read out’, and the corresponding caption of Supplementary Fig. 3.

Comment: “Figure S1c

What is the origin of the crosstalk?”

Reply: We thank the reviewer for raising this important point. The crosstalk originates from the mode overlap between the four angular modes for signal photons. In this work we use angular separation to resolve the four angular-multiplexed (k-vector) signal/idler modes from the same memory cell. The angular separation between the adjacent modes is a couple of times larger than the divergence angle of each mode (the divergence of a Gaussian mode), so that we can pick out each mode without significant efficiency loss (their overlap is small). However, the angular separation is not that large, so there’s still some mode overlap (at the scale of 1%) between adjacent angular modes, which induces some mix and crosstalk between them. The angular separation cannot be too large in this experiment as a large angular separation at the atoms is projected to large spatial separation at the position of the AOD, after the transformation of the three lenses (two of them are 150mm focus, and the other 50mm focus), as shown in the bottom left inset of Supplementary Fig. 1a. Larger spatial separation at the AOD will induce slower switching of the AOD to address different memory cells, which is not favorable. Thus here we find a point in this trade-off between crosstalk and switching time, to achieve a fast switching time of 1.7 μ s and a crosstalk <1% for the experiment.

We have added this discussion in Supplementary Note 1.

Comment: “Figure S1d+e

- a) Specify how the coherence time is measured.
- b) ‘...fitted by retrieval efficiency decay.’ It is not clear what this means.”

Reply: We thank the reviewer for raising this important point. We have replied to a) in the comment regarding line 243 of the main text.

b) We thank the reviewer for raising this important point. The coherence time of the memory can be measured by fitting the decay of the cross correlation or retrieval efficiency. Here we measured with both methods and confirm they yield very similar coherence time. To avoid confusion, we have removed the result fitted by retrieval efficiency in the revised supplementary information, as the coherence time is measured by cross correlation in this experiment.

Reply to the reviewers

Reply to the First Reviewer

Comment: “The manuscript has been largely improved and the new version of the manuscript now provides additional discussions in the main and in the SI. These discussions clarify some of the data and their limitations, providing also insights about future implementations. Given these changes, I can now recommend publication in your journal.”

Reply: We thank the reviewer for the positive comments on the revised manuscript and the recommendation of publication in Nature Communications. We are very thankful for the recommendation of the reviewer.

Reply to the Second Reviewer

Comment: “The authors have addressed most of my comments and questions. In particular, significant revisions have been made regarding the link efficiency, which was pointed out by all reviewers. A quantitative analysis has been conducted on several different setups and parameters for generating entanglement between the remote atoms. However, I still have concerns about their statement that they achieved the link efficiency of 0.92 using the observed rate of 0.46 in their experiment related to L113-116 and L313-319.

In Fig.6, they showed the dependency of the fidelity on the success probability using experimental parameters. However, only from the figure, it is unclear whether the link efficiency larger than 0.83 required for the deterministic delivery of entanglement can be achieved. To clarify their statement, by adding the curve for $\eta=0.83$ on the same figure similar to Fig.1b of Ref.[13], it should be shown that, in the region of $F > 0.5$, there exists a success probability which gives the fidelity of the brown curve (“This work”) larger than that of the curve for $\eta=0.83$. I guess, from the curves of Fig.1b in Ref.[13] and the shape of the brown curve in their manuscript, the fidelity of the brown curve may always be smaller. If correct, I think the link efficiency larger than 0.83 was not achieved using the observed result. The authors should clarify my concern and weaken their argument if needed.

In my opinion, even if the link efficiency larger than 0.83 using the current experimental parameters has not been shown, this paper is worthy of publication after the above concern is addressed, considering their theoretical analysis, which suggests a significant improvement of the link efficiency through an improvement of QFC efficiency likely to be feasible, and the state-of-the-art quality of their experiments including the design, atomic systems and control system.”

Reply: We thank the reviewer for thinking most of the issues have been addressed in the revised manuscript. In the following, let us address the remaining issues point by point. The reviewer is correct that the performance of current setup cannot yield deterministic delivery of remote atom-atom entanglement in the future, and the link efficiency = 0.92 is not realized in this experiment. We have emphasized in the main text that the deterministic entanglement delivery cannot be realized by current memory performance, and weakened the claim of link efficiency, as suggested by the reviewer.

To investigate whether the deterministic entanglement delivery can be achieved, we simulate the fidelity of the deterministic entanglement delivery $F_{det} = p_{succ} * F_{succ} + (1 - p_{succ}) * F_{unent}$ (see Eq. (1) in Ref. [13]) by varying p_{succ} through changing the intrinsic excitation probability χ (p_{succ} is determined by χ) in the revised Fig. 6. We add F_{det} as the secondary vertical axis of Fig. 6, to illustrate all the possible fidelity F_{det} of the deterministic entanglement delivery task. It is noteworthy that χ is the only parameter which can be varied under the 240us protocol of entanglement generation and a fixed memory performance in each curve shown in Fig. 6. This is different from the Fig. 1b in Ref. [13] where the protocol time t_{ent} can be varied (p_{succ} is determined by t_{ent} in Fig. 1b as discussed in the introduction part and the first section of methods in Ref. [13]), but the protocol time t_{ent} is fixed to 240us in our case. It is also noteworthy that, although the curves of F_{succ} versus p_{succ} in Fig. 6 and Fig. 1b in Ref. [13] have very similar appearances, they are based on the variation of different parameters and under different conditions. In each curve in Fig. 1b of Ref. [13], the link efficiency is fixed, but the protocol time t_{ent} is varied. However, in each curve in Fig. 6, the protocol time and the memory performances (such as retrieval efficiency, channel efficiency, memory coherence time) are fixed, but the link efficiency is varied with different p_{succ} . Therefore, it is not feasible to plot a curve with a fixed link efficiency = 0.83 in Fig. 6 as in Fig. 1b of Ref. [13], but the $F_{det} = p_{succ} * F_{succ} + (1 - p_{succ}) * F_{unent}$ for the deterministic entanglement delivery can still be plotted by varying p_{succ} , and the maximally achievable F_{det}^{max} can also be calculated. By plotting the curve of F_{det} in Fig. 6, we find that the highest fidelity for deterministic entanglement delivery task under current memory performance is $F_{det}^{max} = 0.44$, which is lower than the threshold 0.5. Thus we cannot achieve deterministic atom-atom entanglement delivery by combining two setups with current performance, as the reviewer correctly points out. The maximum fidelity $F_{det}^{max} = 0.44$ is achieved at $p_{succ} = 0.57$, corresponding to a link efficiency = 0.79 (calculated by the literal definition: $\eta_{link} = r_{ent}/r_{dec} = T_{coh}/T_{ent}$, i. e., by calculating r_{ent} based on χ , then divided by $r_{dec} = 1/T_{coh}$). On the other hand, with the improvement in frequency conversion in the future, we can achieve $F_{det}^{max} = 0.61$ (see the yellow dashed curve in Fig. 6), which can guarantee the task of deterministic entanglement delivery.

There exist some differences between Fig. 6 and the Fig. 1b in Ref. [13], despite their similar appearances. For example, a link efficiency of 0.79 corresponds to $F_{det}^{max} = 0.49$ according to Eq. (2) and Fig. 1c in Ref. [13], but can only achieve a $F_{det} = 0.44$ in our experiment. The first reason for this discrepancy is that the Fig. 1b and 1c (Ref. [13]) show the ideal case, where the fidelity of the initially generated entanglement is always assumed to be 1 (see first section in methods of Ref. [13]). The link efficiency itself is defined as $\eta_{link} = r_{ent}/r_{dec}$ (see Eq. (1) of this work or the introduction part of Ref. [13]), which can only characterize the infidelity due to decoherence. Other infidelities such as the imperfections in the initially heralded entanglement are not included in link efficiency. Back to our case, except the memory decoherence, we also take other infidelities such as the double excitation error and noise into consideration. Thus the fidelity demonstrated in Fig. 6 is lower but more close to the realistic case. The second reason is that, the protocol time can be arbitrarily varied and set to the optimal value in Fig. 1c (Ref. [13]), but in our case, the duration of the protocol is always fixed to 240us to fulfill the heralding protocol over 12km fiber in our experiment. As we cannot vary the protocol time to optimize F_{det} as in Fig. 1b (Ref. [13]), the achievable F_{det}^{max} is also lower. Therefore, link efficiency = 0.83 demonstrated in Fig. 1c (Ref. [13]) is the lowest link efficiency required for deterministic entanglement delivery in the ideal case. In a realistic case, with inevitable infidelities and restricted protocol, 0.83 cannot guarantee

deterministic entanglement delivery, and a higher link efficiency is required.

In the brown curve in Fig. 6, when we vary the intrinsic excitation probability χ under the fixed memory performance and 240us protocol, the link efficiency also changes linearly with χ . This is guaranteed by the literal definition of link efficiency: $\eta_{link} = r_{ent}/r_{dec}$ (as r_{ent} is linear with χ in single photon interference, and r_{dec} is unchanged). However, higher link efficiency does not necessarily yield higher F_{det} (Fig. 6), as a higher χ means higher double excitation noise which reduces the fidelity of the initially generated entanglement in our case. The lower initial fidelity can induce lower F_{det} , despite a higher link efficiency is achieved in this case. For example, the optimal fidelity of deterministic entanglement delivery task in our current case $F_{det}^{max} = 0.44$ is achieved at a link efficiency = 0.79, not at higher link efficiency with larger χ . Thus there exists a one-to-one correspondence between link efficiency and F_{det}^{max} only in the ideal case as shown in Fig. 1c (Ref. [13]), where initial fidelity is set to 1 and the protocol time can be varied. In a realistic case where additional infidelities and restrictions are considered, this relationship will no longer be valid, and the F_{det}^{max} also depends on other parameters.

As discussed above, the link efficiency is defined as $\eta_{link} = r_{ent}/r_{dec}$ literally. Thus when we combine two atom-photon correlations with link efficiency = η_{a-p} into an atom-atom entanglement in the future, the link efficiency of atom-atom entanglement will be doubled to $\eta_{a-a} = 2\eta_{a-p}$, which is guaranteed by the literal meaning of link efficiency and not related to the task of deterministic entanglement delivery here. This is because the success probability doubles (thus r_{ent} also doubles) when heralding the atom-atom entanglement by single photon interference, but the time cost for each entangling attempt and the memory coherence time are still the same as in the atom-photon entanglement (see Eq. (20-22) in Supplementary Note 11 for details).

As described in the introduction part of Ref. [13], achieving a link efficiency of unity is a rough threshold of quantum network's scale-up. Thus we think achieving a link efficiency of the order of unity is already a significant breakthrough towards the scale-up of a quantum repeater, no matter whether the deterministic entanglement delivery can be achieved, as the reviewer says: "In my opinion, even if the link efficiency larger than 0.83 using the current experimental parameters has not been shown, this paper is worthy of publication after the above concern is addressed, considering their theoretical analysis, which suggests a significant improvement of the link efficiency through an improvement of QFC efficiency likely to be feasible, and the state-of-the-art quality of their experiments including the design, atomic systems and control system." In addition, the performance in this manuscript is not far away from achieving deterministic entanglement delivery in the future. We estimate that an improvement in the efficiency of quantum frequency conversion by roughly 70% can already yield the result of deterministic entanglement delivery, which should be not very hard to achieve in the future.

We have revised the corresponding part in L113-116, L313-319, the caption of Fig. 6 and the paragraph above the discussion section to emphasize that the deterministic entanglement delivery cannot be achieved under current memory performance, and revised the corresponding statements regarding the link efficiency.

Comment: "Other comments:

(1)

In Eq.5 of the supplementary information, $P_i = \chi (1 + \eta(t)) \eta_i$ is used. For $\eta(t)=0$, P_i is not zero. Why? Are any assumptions being made for this equation?"

Reply: We thank the reviewer for raising this important point. The p_i in DLCZ protocol includes two components, the coherent readout of directional emission from spin-wave, and the incoherent noise photon emitted in a 4π solid angle during the readout. The coherent part from spin-wave can be expressed as $\chi\eta(t)\eta_i$, which drops to zero when the intrinsic retrieval efficiency is reduced to zero. However, the incoherent noise $\chi\eta_i$ is always there. At the beginning of the DLCZ protocol, all the atoms are prepared in $|g\rangle$ state by optical pumping. In the write process, a fraction of the atoms are scattered to a metastable state $|s\rangle$ via the excited state $|e\rangle$ by the write pulse, and the scattered photons from $|e\rangle$ to $|s\rangle$ are emitted over a 4π solid angle. Only a very small portion of the scattered photon can be collected into the signal fiber to herald the spinwave-photon entanglement, and the vast majority of the atoms scattered to $|s\rangle$ state are not correlated to the generated spin-wave. These atoms stay in the $|s\rangle$ state until the read process, in which the strong read pulse pumps all the atoms in $|s\rangle$ back to $|g\rangle$. During the read process, the incoherent scattered photons from these atoms are emitted over a solid angle of 4π , and some of the photons can be collected into the idler fiber, which is the incoherent part in p_i . This incoherent scattering will not drop with time due to the metastable level $|s\rangle$ has a very long lifetime. This noise is also proportional to χ as more atoms will be scattered to $|s\rangle$ if higher excitation probability is used. This model is consistent with our observation in the experiment.

Comment: "(2)

For Figs.S11-e and f, the labels of the secondary vertical axes are "Link efficiency."

Are they the link efficiency, not the success probability?

The figures look strange because a larger value of χ gives a smaller value of fidelity. This result implies that the contribution of the success probability improvement is dominant compared with the fidelity decrease because the link efficiency depends on the fidelity and probability. Is my understanding correct?"

Reply: We thank the reviewer for raising this important point. Here the 'Link efficiency' used in the secondary vertical axes is just the literal meaning of link efficiency, which does not correspond to the fidelity of the deterministic entanglement delivery task. As the link efficiency is defined as $\eta_{link} = r_{ent}/r_{dec}$, the link efficiency is linear with the intrinsic excitation probability χ because r_{ent} is linear with χ in single photon heralding and all other parameters such as memory coherence time and protocol time are fixed. The secondary vertical axes can also be replaced by the success probability as pointed out by the reviewer. Note that there is no one-to-one correspondence between the link efficiency and the maximum fidelity F_{det}^{max} in deterministic entanglement delivery in our experiment, and the link efficiency only characterizes the infidelity due to decoherence. Here link efficiency is simply the ratio between the entanglement generation rate and memory decoherence rate in its literal meaning ($\eta_{link} = r_{ent}/r_{dec}$), and not connected to the deterministic entanglement delivery task, as explained in reply to previous comment. To avoid confusion, we have revised the secondary vertical axes to cumulative success probability as suggested by the reviewer.

Here the fidelity decreases with larger intrinsic excitation probability due to larger double excitation error and reduced cross correlation induced by larger intrinsic excitation probability χ , as shown in Supplementary Eq. (10) and (12). These figures illustrate the trade-off between the atom-atom entanglement fidelity and the success probability, as correctly pointed out by the reviewer. However, if the intrinsic excitation probability gets very large, the infidelity due to double excitation will become dominant compared to the improvement in success probability, and a lower F_{det} will be yielded, as shown in the dashed curves in the revised Fig. 6. We have changed the secondary vertical axes from link efficiency to success probability and revised corresponding texts to avoid confusion.

Reply to the Third Reviewer

Comment: “The authors answered all my questions and added additional detailed information in the supplementary material that helps a lot to understand the general picture of a quantum repeater based on this system. The trade-off between entanglement-rate and -fidelity is nicely discussed and the possible steps of improvement included. I recommend its publication.”

Reply: We thank the reviewer for appreciating the revised manuscript and the recommendation of publication in Nature Communications. We are very thankful for the recommendation of the reviewer.